# A small-molecule P2RX7 activator promotes anti-tumor immune responses and sensitizes lung tumor to immunotherapy

Laetitia Douguet[1,15✉], Serena Janho dit Hreich [1,2,3,15], Jonathan Benzaquen[1,2,3], Laetitia Seguin[1,2], Thierry Juhel[1], Xavier Dezitter[4,5], Christophe Duranton [6], Bernhard Ryffel[7], Jean Kanellopoulos[8], Cecile Delarasse [9], Nicolas Renault[4,5], Christophe Furman[4,5], Germain Homerin[4,10], Chloé Féral[1,2], Julien Cherfils-Vicini [1], Régis Millet[4,5], Sahil Adriouch [11], Alina Ghinet [4,10,12], Paul Hofman[1,2,13,14] & Valérie Vouret-Craviari [1,2,3✉]

Only a subpopulation of non-small cell lung cancer (NSCLC) patients responds to immunotherapies, highlighting the urgent need to develop therapeutic strategies to improve patient outcome. We develop a chemical positive modulator (HEI3090) of the purinergic P2RX7 receptor that potentiates αPD-1 treatment to effectively control the growth of lung tumors in transplantable and oncogene-induced mouse models and triggers long lasting antitumor immune responses. Mechanistically, the molecule stimulates dendritic P2RX7-expressing cells to generate IL-18 which leads to the production of IFN-γ by Natural Killer and CD4[+] T cells within tumors. Combined with immune checkpoint inhibitor, the molecule induces a complete tumor regression in 80% of LLC tumor-bearing mice. Cured mice are also protected against tumor re-challenge due to a CD8-dependent protective response. Hence, combination treatment of small-molecule P2RX7 activator followed by immune checkpoint inhibitor represents a strategy that may be active against NSCLC.

[1] Université Côte d'Azur, CNRS, INSERM, IRCAN, Nice, France. [2] FHU OncoAge, Nice, France. [3] Centre Antoine Lacassagne, Nice, France. [4] Inserm, CHU Lille, U1286—Infinite—Institute for Translational Research in Inflammation, University of Lille, Lille, France. [5] Institut de Chimie Pharmaceutique Albert Lespagnol, IFR114, Lille, France. [6] Université Côte d'Azur, CNRS, INSERM, LP2M, Nice, France. [7] INEM—UMR7355, Institute of Molecular Immunology and Neurogenetic, University and CNRS, Orleans, France. [8] Institute for Integrative Biology of the Cell (I2BC), CEA, CNRS, Université Paris-Saclay, Gif-sur-Yvette Cedex, France. [9] INSERM, CNRS, Institut de la Vision, Sorbonne Université, Paris, France. [10] Hautes Etudes d'Ingénieur (HEI), JUNIA, UC Lille, Laboratoire de Chimie Durable et Santé, Lille, France. [11] Institute for Research and Innovation in Biomedicine, Normandie University, Rouen, France. [12] Faculty of Chemistry, 'Al. I. Cuza' University of Iasi, Iasi, Romania. [13] Hospital-Related Biobank (BB-0033-00025), Pasteur Hospital, Nice, France. [14] Laboratory of Clinical and Experimental Pathology and Biobank, Pasteur Hospital, Nice, France. [15]These authors contributed equally: Laetitia Douguet, Serena Janho dit Hreich. ✉email: ldouguet@gmail.com; valerie.vouret@univ-cotedazur.fr

Despite new biological insights and recent therapeutic advances, many tumors remain resistant to treatments, leading to premature death of the patient. This is particularly true for lung cancer, which is the leading cause of cancer death for men and women worldwide. The 5-year survival rate for patients with any type of lung cancer is around 20%, which dramatically drops to 6% for metastatic lung cancers. Recent advances in effective therapies such as targeted therapies and immunotherapies have revolutionized lung cancer treatments[1]. However, it is limited to a small percentage of patients and alternative approaches are urgently needed to improve patient outcome.

The P2RX7 receptor (also called P2X7R) is an ATP-gated ion channel composed of three protein subunits (encoded by the *P2RX7* gene), which is expressed predominantly in immune cells and in some tumor cells[2]. Activation of P2RX7 by high doses of extracellular ATP (eATP) leads to $Na^+$ and $Ca^{2+}$ influx, and, after prolonged activation, to the opening of a larger conductance membrane pore. One consequence of this large pore opening, a unique characteristic of P2RX7, is to induce cell death in eATP rich microenvironments. Noteworthy, such high doses of eATP are present in the inflammatory and tumor microenvironments (TMEs)[3]. P2RX7 functions are largely described in immune cells, where it is involved in NLRP3 activation to induce the maturation and secretion of IL-1β and IL-18 pro-inflammatory cytokines by macrophages and dendritic cells (DCs)[4]. In line, several P2RX7 inhibitors have been developed with the aim to treat inflammatory diseases. In addition to its ability to finely tune the amplitude of the inflammatory response[5], P2RX7 has been shown to orchestrate immunogenic cell death (ICD) and to potentiate DC activation and ability to present tumor antigens to T cells[6]. Among immune cells, regulatory T cells (Treg) are highly sensitive to P2RX7-induced cell death and, in the presence of eATP, P2RX7 negatively regulates their number and their suppressive function[7]. Such response can participate in P2RX7-dependent immune surveillance by unleashing the effector functions of adaptive immune T cells[8]. Therefore, P2RX7 has been proposed to represent a positive modulator of antitumor immune response. This is in agreement with data from our group showing that P2RX7-deficient mice are more sensitive to colitis-associated cancer[9]. Also, in this model, we noticed that transplanted Lewis lung carcinoma (LLC) tumors grew faster in line with the findings of Adinolfi et al. using transplanted B16 melanoma and CT26 colon carcinoma tumors[10]. Collectively, these results support the notion that P2RX7 expression by host immune cells coordinates antitumor immune response.

Capture of tumor antigens by antigen-presenting DC is a key step in immune surveillance. Activated DCs present tumor antigens to naïve T cells leading to their activation and differentiation in effector T cells. Tumor infiltrated effector T cells and NK cells can recognize and kill tumor cells resulting in the release of additional tumor antigens and amplification of the immune response. However, this response is often inhibited by immunosuppressive mechanisms present within the TME. Different mechanisms sustain tumor escape as the reduced immune recognition of tumors due to the absence of tumor antigens, or the loss of MHC-I and related molecules, the increased resistance of tumor cells edited by the immune responses, and the development of a favorable TME associated with the presence of immunosuppressive cytokines and growth factors (such as VEGF, TGF-β) or the expression of checkpoint inhibitors such as PD-1/PD-L1[11]. Inhibitory checkpoint inhibitors (αPD-1/PD-L1 and anti-CTLA-4) are used in daily practice for the treatment of advanced malignancies, including melanoma and non-small cell lung cancer (NSCLC)[12]. These antibodies reduce immunosuppression and reactivate cytotoxic effector cell functions to elicit robust antitumor responses[13,14]. High response rate to αPD-1/PD-L1 therapy is often associated with immune inflamed cancer phenotype characterized by the presence in the TME of both CD4+ and CD8+ T cells, PD-L1 expression on infiltrating immune and tumor cells and many pro-inflammatory and effector cytokines, such as IFN-γ[15]. Noteworthy, only few cancer patients achieve a response with anti-immune checkpoint administered as single-agent[16], suggesting that strategies based on combined therapies would likely enhance antitumor efficacy and immunity.

Despite the role of P2RX7 in stimulating antitumor immunity and the observation that tumor development is more aggressive in *p2rx7*-deficient animals[9], it is currently not known whether P2RX7 activation can modulate tumor progression in vivo. The purpose of this study is to investigate the effect of a positive modulator (PM) of P2RX7 on lung tumor fate. To do so, we use syngeneic immunocompetent tumor mice models and show that activation of P2RX7 improves mice survival. Mechanistically, activation of P2RX7 leads to increased production of IL-18 in a NLRP3-dependent manner, which in turn activates NK and CD4+ T cells to produce IFN-γ and consequently increases tumor immunogenicity. Finally, activation of P2RX7 combined with αPD-1 immune checkpoint inhibitor allows tumor regression, followed by the establishment of a robust immunological memory response.

## Results

**HEI3090 is a positive modulator of P2RX7.** In order to identify positive modulator of P2RX7, 120 compounds from the HEI's proprietary chemical library were screened for their ability to increase P2RX7-mediated intracellular calcium concentration during external ATP exposure. We first produced an HEK cell line expressing the cDNA encoding for *P2RX7* from C57BL/6 origin (HEK mP2RX7) and determined the minimal dose of ATP (333 μM) that should be used to initiate an increase in $Ca^{2+}$ concentration. We tested five promising compounds and identified HEI3090 as a hit (patent WO2019185868A1). HEI3090 corresponds to a pyrrolidin-2-one derivative decorated with a 6-chloropyridin-3-yl-amide in position 1 and with a 2,4-dichlorobenzylamide moiety in position 5 (Fig. 1a and Supplementary Fig. 1). HEI3090 alone showed no toxic activity, was unable to induce intracellular $Ca^{2+}$ variation, and required the presence of eATP to rapidly and dose dependently enhance the P2RX7-mediated intracellular calcium concentration (Fig. 1b, c). The maximum effect of HEI3090 was observed at 250 nM, which is in the range of doses identified in pharmacokinetic analysis (Fig. 1g). HEI3090 action required the expression of P2RX7, since HEK cells transfected with empty plasmid (pcDNA6) showed no increase in intracellular calcium concentration (Fig. 1c). P2RX7 has the unique capacity to form a large pore under eATP stimulation. Large pore opening of P2RX7 was assayed with the quantification of the uptake of the fluorescent TO-PRO-3 dye. As expected, HEI3090 alone had no effect and required eATP stimulation to enhance TO-PRO-3 entry within the cells. HEI3090 increased by 2.5-fold the large pore opening (Fig. 1d). The rapid uptake of TO-PRO-3 was consistent with direct P2RX7 activation rather than ATP/P2RX7-induced cell death (Fig. 1e). We also tested HEI3090's effect on splenocytes expressing physiologic levels of P2RX7 (Fig. 1f). In these immune cells, HEI3090 alone did not affect Fluo-4-AM nor TO-PRO-3 uptake. However, in the presence of eATP, HEI3090 enhanced $Ca^{2+}$ influx and TO-PRO-3 uptake. We also showed that HEI3090 required the expression of P2RX7, since its effect was lost in splenocytes isolated from $p2rx7^{-/-}$ mice.

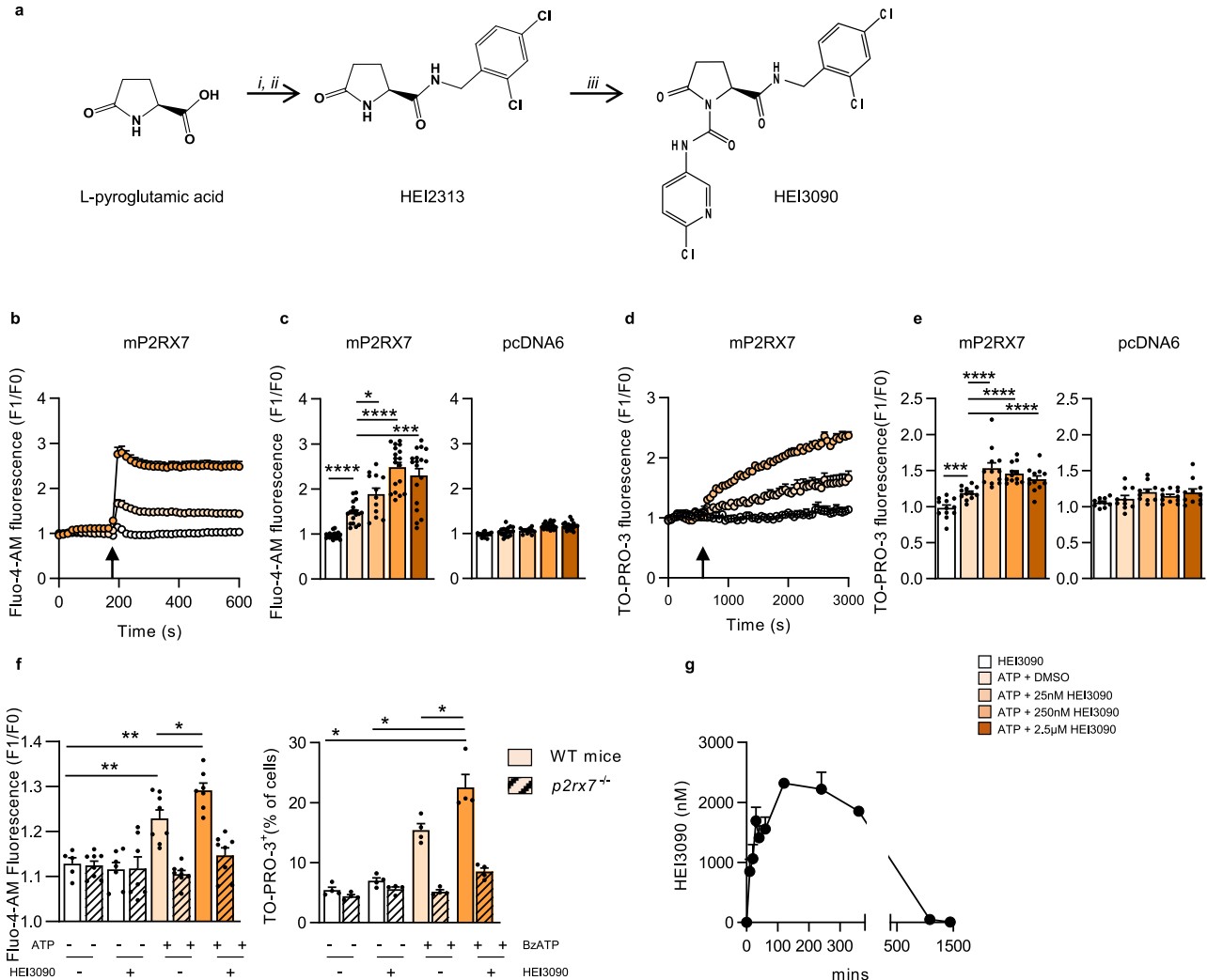

**Fig. 1 HEI3090 enhances ATP-induced receptor channel activity. a** Representation of HEI3090's synthesis steps. **b** Modulation of ATP-induced intracellular $Ca^{2+}$ variation (F1/F0) in HEK293T-mP2RX7 cells (C57Bl/6 origin). After ten baseline cycles, ATP (333 μm) and HEI3090 (250 nM) were injected. Error bars are mean ± SEM ($n = 3$ independent experiments, 6 replicates). **c** Average Fluo-4-AM fluorescence intensities in HEK293T mP2RX7 or control HEK pcDNA6 measured 315 s after stimulation with ATP (333 μM) and HEI3090 at concentrations of 25, 250, and 2.5 μM, as indicated in the color code. Data are presented as scatter dot plots ± SEM ($n = 3$ independent experiments and 6 replicates, two-tailed Mann–Whitney test). **d** Modulation of ATP-induced TO-PRO-3 uptake in HEK293T mP2RX7 cells (F1/F0) in cells treated with ATP and HEI3090 (25 nM). Error bars are mean ± SEM ($n = 3$ independent experiments, 4 replicates, two-tailed Mann–Whitney test). **e** Average fluorescence intensities in HEK293T mP2RX7 or control HEK pcDNA6 measured 10 min after stimulation with ATP and HEI3090 at concentrations of 25, 250, and 2.5 μM, as indicated in the color code. Data are presented as scatter dot plots ± SEM ($n = 3$ independent experiments and 4 replicates, two-tailed Mann–Whitney test). **f** Left: average Fluo-4-AM fluorescence intensities in WT or $p2rx7^{-/-}$ splenocytes stimulated with 50 μM ATP measured at the plateau i.e., 10 min after stimulation. Data are presented as scatter dot plots ± SEM ($n = 2$ independent experiments and 4 replicates). Right: graph represents the percentage of TO-PRO-3 positive cells in splenocytes isolated from naïve WT or $p2rx7^{-/-}$ mice. Data are presented as scatter dot plots ± SEM ($n = 3$ independent experiments in duplicate, two-tailed Man–Whitney test). **g** Pharmacokinetic analysis of HEI3090 intraperitoneally injected in WT mice. Error bars are means ± SEM ($n = 3$ independent experiments in duplicate). Bars are mean ± SEM. $p$ values: *$p < 0.05$, **$p < 0.01$ ***$p < 0.001$, ****$p < 0.0001$. Source data are provided as a Source Data file.

Collectively these results demonstrate that HEI3090 requires P2RX7 expression to be active and enhances eATP-induced P2RX7 activation.

**HEI3090 inhibits tumor growth and enhances antitumor efficacy of αPD-1 treatment**. We previously suggested that P2RX7 expression might favor the activation of immune responses[9]. We therefore evaluated the immuno-stimulatory effect and antitumor efficacy of HEI3090 in vivo, hypothesizing that the high level of eATP contained within the TME[17] would be sufficient to stimulate P2RX7. To do so, we used syngeneic LLC and B16-F10 melanoma cell lines expressing P2RX7 (Supplementary Fig. 2). Vehicle or

HEI3090 (1.5 mg/kg) was administered concomitantly to LLC tumor cell injection and mice were treated daily for 11 days. Mice treated with HEI3090 displayed significantly reduced tumor growth and more than fourfold decrease in tumor weight (Fig. 2a and Supplementary Fig. 3). We next tested the efficacy of HEI3090 to inhibit tumor growth in a therapeutic model, in which treatment started when tumor reached 10–15 mm² in size. HEI3090 (3 mg/kg) inhibited tumor growth and increased by twofold the median survival (Fig. 2b). We also tested the effect of HEI3090 in the melanoma B16-F10 tumor mouse model and observed the same efficacy (Supplementary Fig. 4a, b).

Given the efficacy of HEI3090 to inhibit tumor growth, we next evaluated the combination of HEI3090 and αPD-1 antibody.

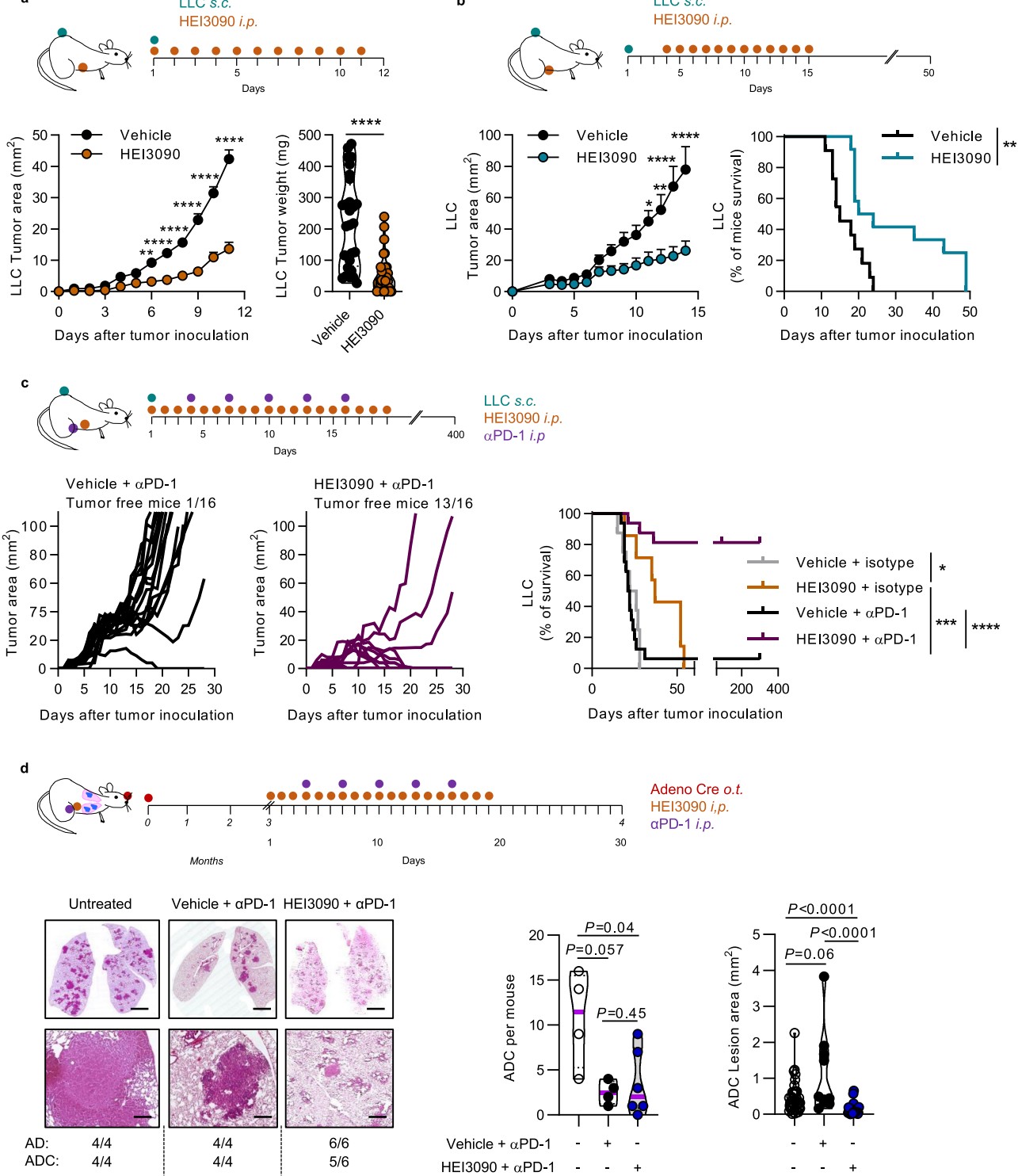

After tumor inoculations, mice were treated daily with HEI3090 or vehicle and αPD-1 was administered at days 4, 7, 10, 13, and 16. While only 1 mouse out of the 16 mice treated with the αPD-1 alone showed a tumor regression, 13 out of the 16 mice treated with HEI3090 + αPD-1 were tumor-free, suggesting that this molecule increased the efficacy of immune checkpoint inhibitor to induce effective antitumor immune responses and tumor regression (Fig. 2c). Importantly, only the combo treatment allows a long-lasting improved survival of at least 340 days (Fig. 2c, right panel). The combo treatment also increased the survival of mice grafted with B16-F10 tumors (Supplementary

Fig. 4c). As illustrated in Fig. 2d, we tested the combo treatment on the KRAS-driven lung cancer (LSL $Kras^{G12D}$) model, which leads to adenocarcinomas 4 months after instillation of adenoviruses expressing the Cre recombinase[18]. Whereas αPD-1 treatment tends to reduce the number of ADC (Fig. 2d), HEI3090 was able to enhance αPD-1's effects in this mouse model. Indeed, tumor burden in mice treated with the combo treatment is reduced by 60% compared to mice treated with αPD-1 alone. Accordingly, the cell number per $mm^2$ and the number of cells positively stained for the proliferation marker Ki67 were decreased by 50% in lesion areas (Supplementary Fig. 4d). One

**Fig. 2 HEI3090 inhibits tumor growth and combined with immunotherapy ameliorates mice survival. a** Prophylactic administration. Average tumor area and weight of LLC allograft after daily treatment with HEI3090. Curves showed mean tumor area in mm$^2$ ± SEM (vehicle $n = 28$, HEI3090 $n = 32$, two-way Anova test, left panel) and graph showed tumor weight the day of sacrifice. Data are presented by violin plots showing all points with hatched bar corresponding to median tumor weight (vehicle $n = 28$, HEI3090 $n = 32$. Two-tailed Mann–Whitney test, right panel). **b** Therapeutic administration. Average tumor area and survival curves of LLC allograft. HEI3090 started when tumors reached 10–15 mm$^2$ of size. Curves showed mean tumor area in mm$^2$ ± SEM ($n = 12$, two-way Anova test, left panel and Mantel Cox test, right panel). **c** Combo treatment. Average tumor area of LLC allograft after HEI3090 and αPD-1 treatment. Spaghetti plots and survival curves of animals are shown ($n = 16$, Mantel Cox test). **d** Schematic illustration of treatment given to LSL-Kras$^{G12D}$ mice. Representative images showing lung tumor burden (Bar = 2 mm upper panel and 500 µm lower panel) with tumor histopathology ($n = 4$). Average tumor burden of LSL-KrasG12D mice in response to treatments were studied as the number of ADC per mouse (untreated, $n = 4$, vehicle + αPD-1, $n = 4$, HEI3090 + αPD-1, $n = 6$; two-tailed Mann–Whitney test) and the surface of ADC lesions per lung. Each point represents one lesion, all lesions are shown to illustrate the heterogeneity (untreated and vehicle + αPD-1, $n = 4$ mice, HEI3090 + αPD-1, $n = 6$, Two-tailed Mann–Whitney test). $p$ values: *$p < 0.05$, **$p < 0.01$ ***$p < 0.001$, ****$p < 0.0001$. Source data are provided as a Source Data file. AD adenoma, ADC adenocarcinoma.

mouse out of the six treated with HEI3090 and αPD-1 was protected against adenocarcinoma formation.

**DCs mediate the antitumor effect of HEI3090.** LLC tumor cells express an active P2RX7, since the presence of high doses of eATP leads to an increase in intracellular Ca$^{2+}$ concentration, which is blocked by the GSK1370319A P2RX7 inhibitor[19] (Supplementary Fig. 2a). To functionally investigate which cells are targeted by HEI3090, we inoculated LLC in $p2rx7^{-/-}$ mice and treated them with HEI3090. Whereas HEI3090 efficiently blocked LLC tumor growth in WT mice (Fig. 2a), the same treatment was inefficient in $p2rx7^{-/-}$ mice, as tumor growth was indistinguishable in treated or untreated groups (Fig. 3a). This result suggests that HEI3090 requires P2RX7 expression by mouse host cells to inhibit tumor growth. The importance of immune cells was further confirmed by the demonstration that the antitumor efficacy of HEI3090 was restored after adoptive transfer of WT splenocytes into $p2rx7^{-/-}$ mice. DCs express P2RX7 and orchestrate antitumor immunity. Purified DC from WT spleens transferred into $p2rx7^{-/-}$ mice were able to restore the antitumor effect of HEI3090 (Fig. 3b). This experiment was further supported by the fact that phagocytic cells (macrophages and DC) were required for HEI3090's antitumor effect (Supplementary Fig. 5a) and that macrophages are less implicated in HEI3090's effect in vivo since HEI3090 is still able to inhibit tumor growth in $p2rx7^{fl/fl}$ LysM mice (Supplementary Fig. 5b).

Flow cytometry analyses revealed that the TME of mice treated with HEI3090 were more infiltrated by immune cells than control mice (Fig. 3d). An increased infiltration of CD8$^+$ T cells was also observed in the LSL-Kras$^{G12D}$ lung tumor mouse model (Fig. 3e). Furthermore, we showed that HEI3090-treated mice showed higher levels of P2RX7 on DC (Supplementary Fig. 5c). Deeper characterization of immune cell infiltrate in the LLC tumor model revealed that anti-CD3 staining of tumors from HEI3090-treated mice contained four times more CD3$^+$ T cells than tumors from vehicle-treated mice (Fig. 3f). Whereas the proportion of CD4$^+$FOXP3$^+$ Treg cells was comparable between treated or untreated mice (Fig. 3g), we found fewer myeloid derived suppressor cells (PMN-MDSCs) after HEI3090 therapy (Fig. 3h) and higher NK/PMN-MDSC and CD4/PMN-MDSC ratios (Fig. 3i) but the treatment failed to consistently increase the CD8/PMN-MDSC ratio. We also showed that HEI3090 targets immune cells in the low immunogenic B16-F10 melanoma syngeneic mouse model, where it was able to increase antitumor effector cells and decrease M-MDSCs infiltration (Supplementary Fig. 6).

P2RX7 expressed by DC has been shown to link innate and adaptive immune responses against dying tumor cells upon chemotherapy-induced ICD and facilitate tumor antigens presentation to T cells[6]. We evaluated the capacity of HEI3090

treatment to kill tumor cells and concomitant stimulation of DC maturation. Our results showed that HEI3090 is not an ICD inducer (Supplementary Fig. 7).

The two tumor cell lines used in this study express different levels of P2RX7 (Supplementary Fig. 2), yet HEI3090 required P2RX7's expressing immune cells to inhibit tumor growth in both tumor mouse models. These results demonstrate that HEI3090 controls tumor growth by recruiting and activating P2RX7-expressing immune cells, especially DC, within the TME to initiate an effective antitumor immune response.

**IL-18 is produced in response to HEI3090 treatment and is required to mediate its antitumor activity.** We then investigated how the activation of P2RX7 enhanced antitumor immune responses. In addition to increasing intracellular Ca$^{2+}$ concentration and stimulating the formation of a large membrane pore (see Fig. 1), P2RX7's activation is also known to activate the NLRP3 inflammasome that leads to the activation of caspase-1 and consequently to the maturation and release of the pro-inflammatory cytokines IL-1β and IL-18. We showed that HEI3090 enhanced caspase-1 cleavage (Supplementary Fig. 8). Whereas neutralization of IL-1β did not impact HEI3090's antitumor activity, neutralization of IL-18 suppressed the antitumor effect of HEI3090 (Fig. 4a). This result was confirmed using $il18^{-/-}$ mice in which HEI3090 had no impact on tumor growth (Fig. 4b). IHC staining of LLC tumors from HEI3090-treated mice showed a significant intratumor amount of IL-18 compared to mice treated with the vehicle (Fig. 4c), whereas staining of tumors from $il18^{-/-}$ mice revealed no staining (Supplementary Fig. 9a). Concordantly, serum levels of IL-18 were statistically more abundant in mice treated with HEI3090 than in vehicle mice (Fig. 4d), and no IL-18 was detected in the serum of mice that received IL-18 neutralizing antibody. In addition, HEI3090 was unable to modulate the levels of IL-18 in $p2rx7^{-/-}$ mice. Moreover, HEI3090-treated WT and $p2rx7^{-/-}$ mice show indeed a significant difference in the release of IL-18. Finally, primary peritoneal macrophages and bone-marrow-derived dendritic cells (BMDC) from WT mice-cultured ex vivo with ATP and HEI3090 produce more IL-18 than cells cultured with ATP and vehicle (Fig. 4e and Supplementary Fig. 10c). IL-18 release by HEI3090 required the NLRP3 inflammasome, since its production is inhibited by the NLRP3 inflammasome-specific inhibitor (MCC950) (Fig. 4e). Moreover, we showed that HEI3090 enhanced caspase-1 cleavage (Supplementary Fig. 8) meaning that HEI3090 was able to increase IL-18 production by enhancing the activation of the NLRP3 inflammasome. Activation of P2RX7 by HEI3090 in macrophages from $p2rx7^{-/-}$ mice failed to increase IL-18 secretion (Supplementary Fig. 9a) and no staining was observed in LLC tumors from HEI3090-treated $p2rx7^{-/-}$ mice (Supplementary Fig. 9b). In agreement with the observation

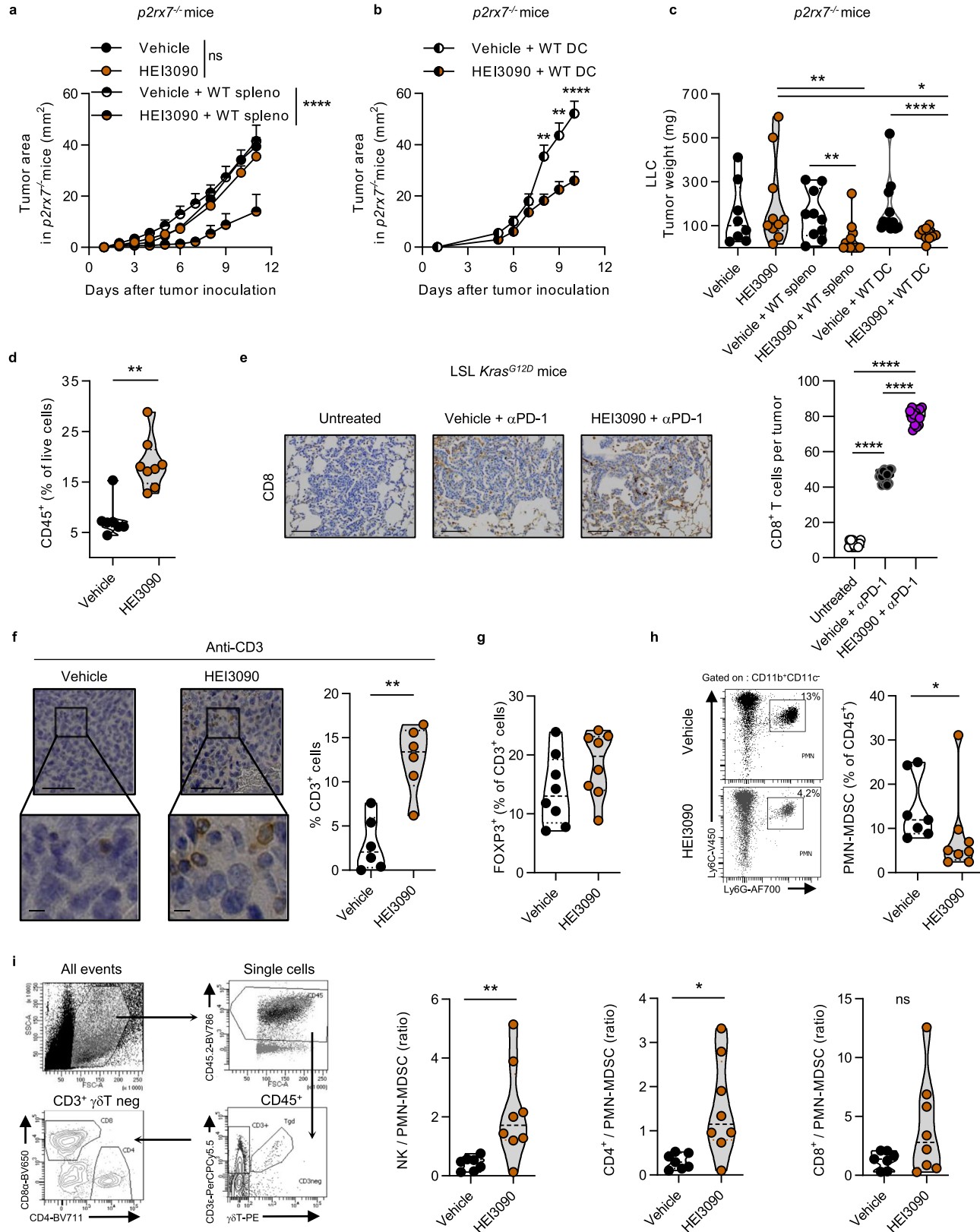

that HEI3090 retained its antitumor activity in mice treated with IL-1β neutralizing antibody, HEI3090 did not modify IL-1β protein levels in serum (Fig. 4d) and did not modulate IL-1β secretion in macrophages cultured ex vivo (Supplementary Fig. 9c, d).

Increased production of IL-18 was also observed in the LSL-Kras$^{G12D}$ lung tumor mouse model. Indeed, cells within lesions of mice treated with HEI3090 combined with αPD-1 expressed more IL-18 than mice treated with αPD-1 alone (Fig. 4f). IL-18 protein levels in serum of mice that received the combo treatment were

**Fig. 3 Immune cells mediate the antitumor activity induced by HEI3090. a** Average tumor area of LLC allograft in *p2rx7*-deficient mice (*p2rx7*$^{-/-}$) after daily treatment with HEI3090 or after adoptive transfer of WT splenocytes and daily treatment with HEI3090. Curves showed mean tumor area in mm$^2$ ± SEM (vehicle $n = 13$, HEI3090 $n = 16$ mice, vehicle + WT spleno $n = 11$, HEI3090 + WT spleno: $n = 12$, two-way Anova test). **b** Average tumor area of LLC allograft in *p2rx7*-deficient mice (*p2rx7*$^{-/-}$) after adoptive transfer of WT DCs and daily treatment with HEI3090. Curves showed mean tumor area in mm$^2$ ± SEM ($n = 8$, two-way Anova test). **c** Tumor weight of animals from the study shown in **a** and **b**. Data are presented by violin plots showing all points with hatched bar corresponding to median tumor weight (vehicle $n = 8$, HEI3090 $n = 10$ mice, vehicle + WT spleno $n = 10$, HEI3090 + WT spleno $n = 12$, vehicle + WT DC $n = 13$, HEI3090 + WT DC $n = 12$ mice, two-tailed Mann–Whitney test). **d** Characterization of immune infiltrate at day 12. Percentage of CD45$^+$ analyzed by flow cytometry among living cells within TME. Data are presented by violin plots showing all points with hatched bar corresponding to median tumor weight (vehicle $n = 7$, HEI3090 $n = 8$, two-tailed Mann–Whitney test). **e** Representative picture of CD8$^+$ cells recruitment in LSL-*Kras*G12D mice over six mice studied (bar = 100 μm) and quantification. Data are presented by violin plots showing all points with hatched bar corresponding to median CD8$^+$ T cells (four tumors per mouse, $n = 4$ mice per group, two-tailed Mann–Whitney test). **f** Representative images of CD3 staining in LLC tumors over six mice studied (bar = 100-μm upper panel and 50-μm lower panel) and quantification data are presented by violin plots showing all points with hatched bar corresponding to median CD8$^+$ T cells ($n = 6$ per group, two-tailed Mann–Whitney test). **g** Percentage of regulatory T cells determined by flow cytometry as FOXP3$^+$ CD4$^+$ among CD3$^+$ within LLC tumors. Data are presented by violin plots showing all points with hatched bar corresponding to median FOXP3$^+$ cells of CD3 cells ($n = 8$ per group, two-tailed Mann–Whitney test). Gating strategy is presented in Supplementary Fig. 12. **h** Proportion of PMN-MDSC among CD45$^+$ within LLC tumors. Data are presented by violin plots showing all points with hatched bar corresponding to median PMN-MDSC cells among CD45$^+$ cells ($n = 7$, per group. Two-tailed Mann–Whitney test). Full gating strategy is presented in Supplementary Fig. 12. **i** Gating strategy (left panel) and ratio of NK, CD4$^+$, or CD8$^+$ T cells on PMN-MDSC within LLC tumors (right panel). Data are presented by violin plots showing all points with hatched bar corresponding to median of indicated cells ($n = 8$ per group, Two-tailed Mann–Whitney test). *p* values: *$p < 0.05$, **$p < 0.01$, ****$p < 0.0001$. Source data are provided as a Source Data file. Spleno Splenocytes, DC dendritic cells, PMN-MDSC poly morpho nuclear-myeloid-derived suppressor cells.

also increased by sixfold (Fig. 4f and Supplementary Fig. 9e). As described with the LLC tumor model, HEI3090 did not impact the levels of IL-1β in this in situ genetic tumor mouse model (Supplementary Fig. 9f).

Collectively, these results demonstrate that the antitumor effect of HEI3090 is highly dependent on P2RX7 expression and on its capacity to induce the production of mature IL-18 in the presence of eATP.

**IL-18 is required to increase antitumor functions of NK and CD4$^+$ T cells**. To identify which immune cells were involved in the HEI3090-induced antitumor response, we performed antibody-specific cell depletion experiments. While NK and CD4$^+$ T cells depletions prevented HEI3090 treatment from inhibiting tumor growth (Fig. 5a, b), CD8$^+$ T cells depletion had no impact on HEI3090 treatment efficacy (Fig. 5a). To further study the effect of HEI3090 treatment on these subsets, we assessed their cytokine production within the TME. Analyses of tumor infiltrating immune cells first revealed that HEI3090 treatment significantly increased their capacity to produce IFN-γ (Fig. 5c). To precisely evaluate which cells in the TME produce IFN-γ, we studied the TIL subpopulation and determined the ratios of IFN-γ to IL-10 production in each subset (Fig. 5d). NK and CD4$^+$ T cells were more biased to produce IFN-γ than the IL-10 immunosuppressive cytokine. CD8$^+$ T cells were relatively less prone to modification in this cytokine ratio profile upon HEI3090 treatment. In addition, twofold more NK cells from mice treated with HEI3090 degranulate after ex vivo restimulation with LLC compared to NK from control mice (Fig. 5e), confirming their activation state, while no effect was noticeable on CD8$^+$ T cells. These phenotypic and functional analyses of intratumor immune infiltration suggested furthermore that treatment with HEI3090 stimulates CD4$^+$ T cells and NK cells' activation in the TME. Importantly, IL-18 neutralization abrogated the increase of the IFN-γ/IL-10 ratio by CD4$^+$ T cells and NK cells (Fig. 5f), suggesting that its production is a direct consequence of IL-18 release and signaling. We showed that DC and IL-18 were necessary for HEI3090's activity (Figs. 3b and 4a, b). In vitro stimulation of splenocytes treated with BzATP and HEI3090 did not increase IFN-γ production by T cells and NK cells indicating that its higher production in the tumor of treated mice is rather an indirect consequence of the therapy

(Supplementary Fig. 10a). Concordantly, CD45$^+$ cells, CD8$^+$ T cells, and NK cells in the TME of *p2rx7*$^{-/-}$ mice supplemented with WT DC showed an increase in the IFN-γ/IL-10 ratio in the HEI3090-treated mice (Supplementary Fig. 10b). This result indicates that WT DC were able to produce IL-18 after the adoptive transfer, since antitumor effector cells were more prone to produce IFN-γ than the IL-10 immunosuppressive cytokine.

Finally, we uncovered that HEI3090 treatment of LLC tumor bearing mice in vivo increased the expression of MHC-I and PD-L1 by 2.2-fold (Fig. 5g). However, when LLC cells were treated in vitro with HEI3090, neither MHC-I nor PD-L1 expression were increased. By contrast, IFN-γ induced the expression of these two proteins (Supplementary Fig. 10d). Taken together, our results suggest that the in vivo increase of MHC-I and PD-L1 expression is a consequence of IFN-γ upregulation driven by IL-18. Finally, using the LSL-*Kras*$^{G12D}$ tumor mouse model, we showed that tumor cells from mice that received both HEI3090 and αPD-1 expressed more PD-L1 than tumor cells from mice treated with αPD-1 only (Fig. 5h). Altogether, our results indicate that HEI3090 increases IL-18 production allowing the recruitment and activation of NK and CD4$^+$ T cells and the production of IFN-γ. In turn, IFN-γ stimulates expression of MHC-I and PD-L1 on cancer cells, leading to an increased-tumor immunogenicity and an increased sensitivity to anti-immune checkpoint inhibitors.

**Combined with αPD-1 antibody, HEI3090 cures mice carrying LLC tumors and allows memory immune response**. Combined with an αPD-1 antibody, HEI3090 cured 80% of LLC-tumor-bearing mice (Fig. 2d). To determine whether cured mice developed an antitumor immune memory response, they were rechallenged with LLC tumor cells 90 days after the first inoculation and were maintained without any therapy as illustrated in Fig. 6a. All long-term-recovered mice were protected from LLC rechallenge, whereas all age-matched control mice developed tumors (Fig. 6b). The rechallenged mice were still alive 150 days after the initial challenge (Fig. 6c), sustaining the hypothesis that combo treatment effectively promoted an efficient antitumor memory immune response. Our results suggested that CD8$^+$ T cells are not directly involved in the primary antitumor effect of HEI3090 (see Fig. 5a). Nevertheless, it is well-characterized that these cells play a pivotal role in the host's ability to mount an

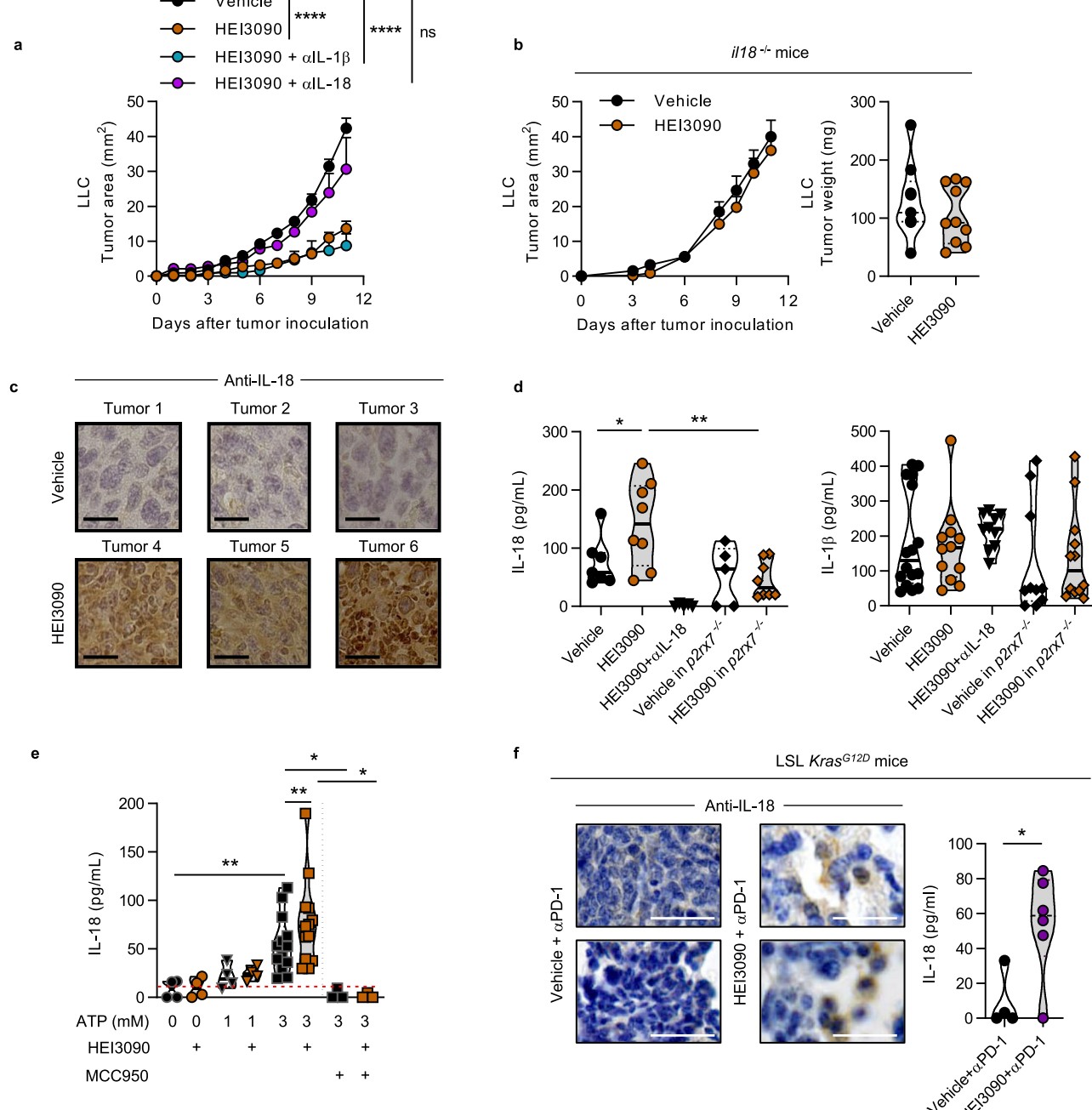

**Fig. 4 HEI3090-induced IL-18 production is required to inhibit tumor growth. a** Average tumor area of LLC allograft in WT mice injected with IL-1β and IL-18 neutralizing antibodies and daily treatment with HEI3090. Curves showed mean tumor area in mm² ± SEM (vehicle n = 28, HEI3090 n = 32, HEI3090 + IL-1β treated n = 6, HEI3090 + IL-18 n = 8. Two-way Anova test). **b** Average tumor area and tumor weight of LLC allograft in *il-18*-deficient mice (*il-18*⁻/⁻) and daily treatment with HEI3090. Curves showed mean tumor area in mm² ± SEM (vehicle n = 9, HEI3090 n = 10. Two-way Anova test, left panel) and graph showed tumor weight the day of sacrifice. Data are presented by violin plots showing all points with hatched bar corresponding to median tumor weight (vehicle n = 9, HEI3090 n = 10. Two-tailed Mann–Whitney test, right panel). **c** Representative images of IL-18 staining in LLC tumors of six mice studied. Bar = 50 μm. **d** Production of IL-18 and IL-1β in serum of treated mice determined by ELISA. Data are presented by violin plots showing all points with plain bar corresponding to median cytokine concentration (IL-18 production: vehicle n = 7, HEI3090 n = 8, HEI3090 + αIL-18 n = 6, vehicle *p2rx7*⁻/⁻ n = 5, HEI3090 *p2rx7*⁻/⁻ n = 8, (IL-1β production: vehicle n = 13, HEI3090 n = 12, HEI3090 + αIL-18 n = 10, vehicle *p2rx7*⁻/⁻ n = 10, HEI3090 *p2rx7*⁻/⁻ n = 12. Two-tailed Mann–Whitney test). **e** Ex vivo production of IL-18 in primary peritoneal macrophages. Data are presented by violin plots showing all points with hatched bar corresponding to median cytokine concentration (no treatment n = 4, HEI3090 n = 4, ATP 1 mM n = 4, ATP 1 mM + HEI3090 n = 4, ATP 3 mM n = 13, ATP 3 mM + HEI3090 n = 13, ATP 3 mM + MCC950 n = 3, ATP 3 mM + HEI3090 + MCC950 n = 3. Two-tailed Mann–Whitney test). **f** Representative images of IL-18 staining in lung tumor lesions from LSL-*Kras*^G12D mice over four mice studied (bar = 100 μm) and production of IL-18 in serum of LSL-*Kras*^G12D mice. Data are presented by violin plots showing all points with hatched bar corresponding to median IL-18 concentration (vehicle + αPD-1, n = 4, HEI3090 + αPD-1, n = 6. Two-tailed Mann–Whitney test). *p* values: *p < 0.05, **p < 0.01, ****p < 0.0001. Source data are provided as a Source Data file.

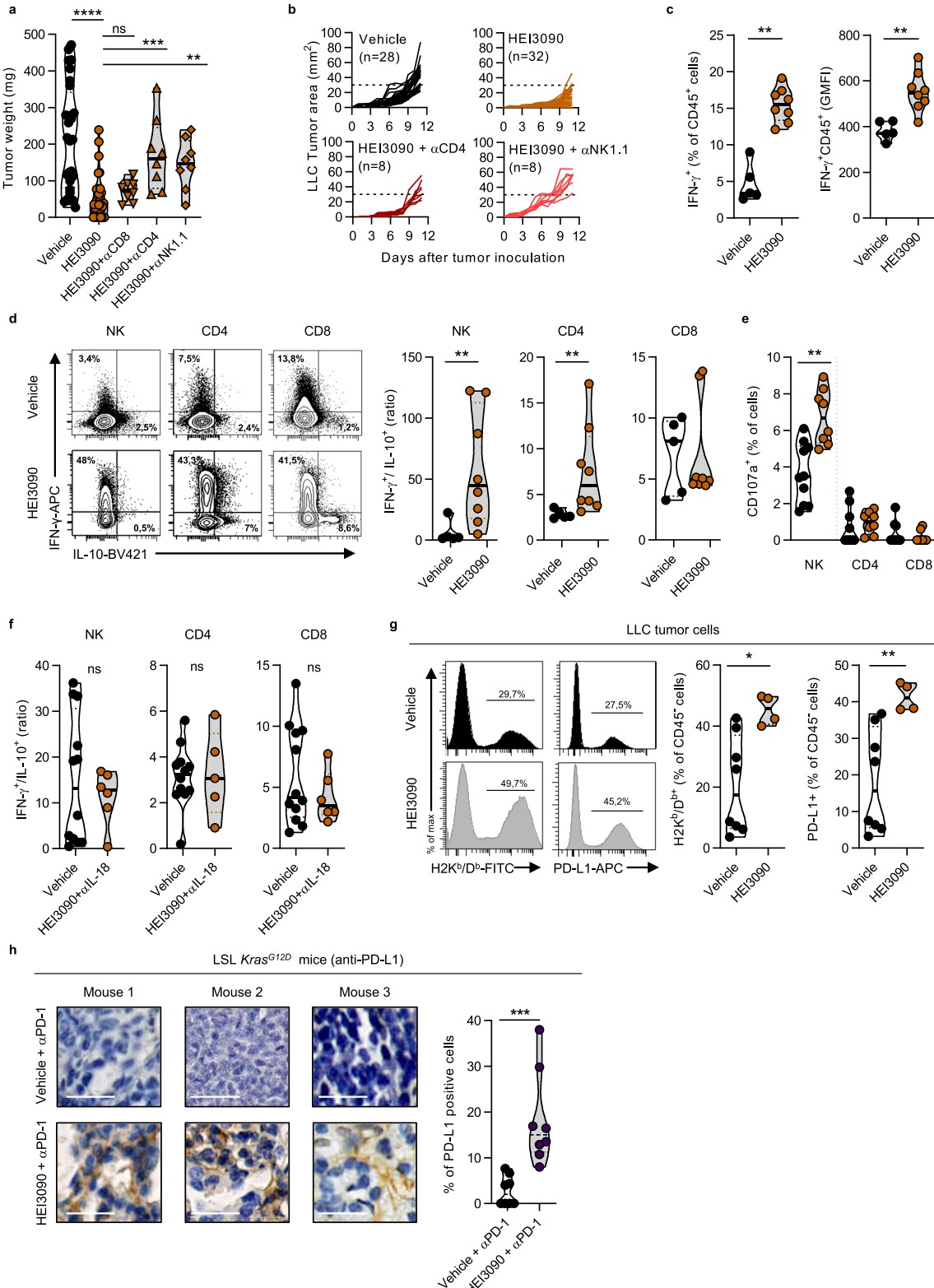

antitumoral adaptative immune response[20]. To evaluate the involvement of secondary memory CD8[+] T cells response in these mice, we sorted CD8[+] cells from age-matched naïve mice or 5 months (day 150) surviving rechallenged mice (see Fig. 6a) and injected them to naïve mice prior to inoculation of LLC tumor cell in a 1/1 ratio. No treatment was given to mice. In this experimental condition, tumor growth was reduced by twofold in mice that received CD8[+] T cells isolated from cured mice

(Fig. 6d), indicating that the combo therapy promoted a functional immune memory response that partly depends on CD8[+] T cells.

We next characterized the mice that were cured for a very long period (300 days), as illustrated in Fig. 6e. First, to discriminate between dormancy and eradication of tumor cells, we depleted CD8[+] T cells from 300-day-old cured mice and followed mice welfare in the absence of treatment (Fig. 6f). In this condition, no

**Fig. 5 HEI3090 triggers antitumor responses mediated by IL-18-induced NK and CD4⁺ T cells. a** Average tumor weight of LLC allograft in WT mice injected with depleting antibody and daily treated with HEI3090. Data are presented by violin plots showing all points with plain bar corresponding to median tumor weight (vehicle $n = 28$, HEI3090 $n = 32$, HEI3090 + αCD8 n = 8, HEI3090 + αCD4 $n = 8$, HEI3090 + αNK1.1 $n = 8$. Two-tailed Mann–Whitney test). **b** Spaghetti plots of LLC allograft in WT mice injected with depleting antibody and daily treated with HEI3090. vehicle $n = 28$, HEI3090 $n = 32$, HEI3090 + αCD4 $n = 8$, HEI3090 + αNK1.1 $n = 8$. **c** Average of IFN-γ⁺ cells among CD45⁺ cells in LLC tumors. Data are presented by violin plots showing all points with plain bar corresponding to median of IFN-γ⁺ cells among CD45⁺ cells (vehicle $n = 5$, HEI3090 $n = 8$. Two-tailed Mann–Whitney test). **d** Representative dot plots of IFN-γ and IL-10 staining on TILs (left panel) and ratios of IFN-γ on IL-10 in the same positive cells of each TILs (right panel). Data are presented by violin plots showing all points with plain bar corresponding to median of the cytokine ratio vehicle $n = 5$, HEI3090 $n = 8$. Two-tailed Mann–Whitney test). **e** Ex vivo degranulation assay of splenocytes from LLC tumor bearing mice. CD107a⁺ cells in NK, CD4⁺, and CD8⁺ T cells are shown. Data are presented by violin plots showing all points with plain bar corresponding to median % of CD107a⁺ cells (vehicle $n = 10$ vehicle and HEI3090 $n = 8$. Two-tailed Mann–Whitney test). **f** Ratios of IFN-γ on IL-10 in the same positive cells of each TILs of IL-18 neutralized mice. Data are presented by violin plots showing all points with plain bar corresponding to median of the cytokine ratio (vehicle $n = 12$, HEI3090 $n = 6$. Two-tailed Mann–Whitney test). **g** Flow cytometry analyses of MHC-I and PD-L1 expression on CD45⁻ cells in LLC tumors. Data are presented by violin plots showing all points with plain bar corresponding to median of positive cells over CD45 cells (vehicle $n = 8$, HEI3090 $n = 4$. Two-tailed Mann–Whitney test). **h** Representative images of PD-L1 staining in cancer lesion of LSL-KRas$^{G12D}$ mice representative of six mice studied. (Bar = 100 μm) and quantification. Data are presented by violin plots showing all points with hatched bar corresponding to median of positive cells over total cells (vehicle $n = 7$, HEI3090 $n = 8$. Two-tailed Mann–Whitney test). $p$ values: *$p < 0.05$, **$p < 0.01$ ***$p < 0.001$, ****$p < 0.0001$. Source data are provided as a Source Data file.

tumor relapse was observed during the 40 days of the experiment and the weight of the mice remained constant, revealing that the combo treatment efficiently eliminated tumor cells. Second, since circulating CD8⁺ T cells are actively involved in the immune memory response[20] and participated in the HEI3090-induced antitumor response (see Fig. 6d), we investigated their involvement in the long-term memory immune response. To do so, 340-day-old cured or age-matched naïve mice were inoculated with LLC tumor cells in the absence of CD8⁺ T cells. Both naïve- and cured-age-matched mice developed tumors (Figs. 6g, h). However, the tumor growth was significantly reduced in cured mice and three out of the six mice did not have tumors (Fig. 6g, right panel). Cured mice survival was also significantly increased in comparison to naïve mice (Fig. 6i). Collectively, these results suggest that circulating CD8⁺ T cells participate in the antitumor immune response induced by HEI3090.

**P2RX7 is positively correlated with high infiltration of antitumor immune cells in NSCLC patients.** Using the lung adenocarcinomas (LUAD) TCGA dataset, we analyzed the effect of *P2RX7* expression levels on the recruitment of cytotoxic immune cells. We clustered tumors of 80 patients with all stage (I-IV) of lung adenocarcinoma according to *P2RX7* expression and showed that high levels of *P2RX7* expression correlated with an increased immune response in LUAD patients, characterized by a high mRNA expression of *CD274* (*PD-L1*), *IL1B*, *IL18*, a signature of primed cytotoxic T cells (defined by *CD8A*, *CD8B*, *IFN-G*, *GZMA*, *GZMB*, *PRF1*) (Fig. 7a). Accordingly, Gene set enrichment analysis (GSEA) demonstrated a positive correlation between high *P2RX7* expression and the well-characterized established signatures of "adaptive immune response," "T-cell-mediated immunity," "cytokine production" (Fig. 7b). Furthermore, high *P2RX7* expression is correlated with high levels of *CD274* (*PD-L1*), independently of the stage of the disease (Fig. 7c). Consistently, a significant reduced overall survival is observed for *P2RX7* hi, *CD274* hi, and *P2RX7* hi + *CD274* hi LUAD patients (Fig. 7d), suggesting that high expression levels of *P2RX7* is sufficient to bypass immune responses in the presence of high levels of *CD274*. Such a situation is considered to benefit from anti-checkpoint blockade and/or strategies aiming to reactivate immune responses, e.g., with an activator of P2RX7. Indeed, only few cancer patients achieve a response with anti-immune checkpoint administered as single-agent and combined therapies to enhance antitumor immunity and bring a clinical benefit for patients are actively tested. We showed in this study

that the combination of HEI3090 and αPD-1 is more efficient to inhibit lung tumor growth than αPD-1 alone (see Fig. 2c).

**Discussion**
We demonstrated in this study that activation of the purinergic P2RX7 receptor represents a promising strategy to control tumor growth. We developed a positive modulator of P2RX7, called HEI3090, that stimulates antitumor immunity. HEI3090 induces production of IL-18 by P2RX7-expressing immune cells, by mainly targeting DC. IL-18 drives IFN-γ production to increase tumor immunogenicity and reinforces NK and CD4⁺ T cells immune responses and generates protective CD8⁺ T cells responses from recidivism. Noteworthy, therapeutic association of HEI3090 with αPD-1 antibody synergizes to cure mice in the LLC syngeneic model of lung cancer and elicits an antitumor immunity. We also observed that the combo treatment is more efficient than αPD-1 alone to inhibit tumor growth in the LSL-KRas$^{G12D}$ lung tumor genetic mouse model. Lung tumor regression correlates with an increased immune cell infiltration, more secretion of IL-18 within the TME and higher expression of PD-L1 by tumor cells. Furthermore, this mode of action was confirmed using the B16-F10 melanoma tumor model (Supplementary Fig. 6). Collectively these results demonstrate that the antitumor activity of HEI3090 follows the same rules in all tumor models tested and highlight the strength of HEI3090 to reactivate antitumor immunity.

The design of P2RX7's modulators was based on a ligand-based approach allowing the generation of a pharmacophore model. One hundred and twenty compounds were generated and were tested for their ability to enhance P2RX7's activities; five of them were able to do so. HEI3090 was the most promising and effective compound of the five and was therefore chosen for our study. Other natural or synthetic molecules have been described to facilitate P2RX7 response to ATP[21–23]. P2RX4 is another member of the P2X family that is described to regulate P2RX7's activities in macrophages. Recently Kawano et al. have shown that a positive modulator of P2RX4, the ginsenoside CK compound[24], calibrates P2RX7-dependent cell death in macrophages[25]. Therefore, we checked whether HEI3090 modulates P2RX4's activities, which is not the case (Supplementary Fig. 11).

Until now, neither of these molecules has been tested in cancer models. Moreover, attempt to facilitate P2RX7 activation in the field of oncology has been limited by the finding that P2RX7 variants expressed by some tumor cells may sustain their proliferation and metabolic activity[2]. To explore this question, we analyzed P2RX7's functional features in ex vivo lung cancer

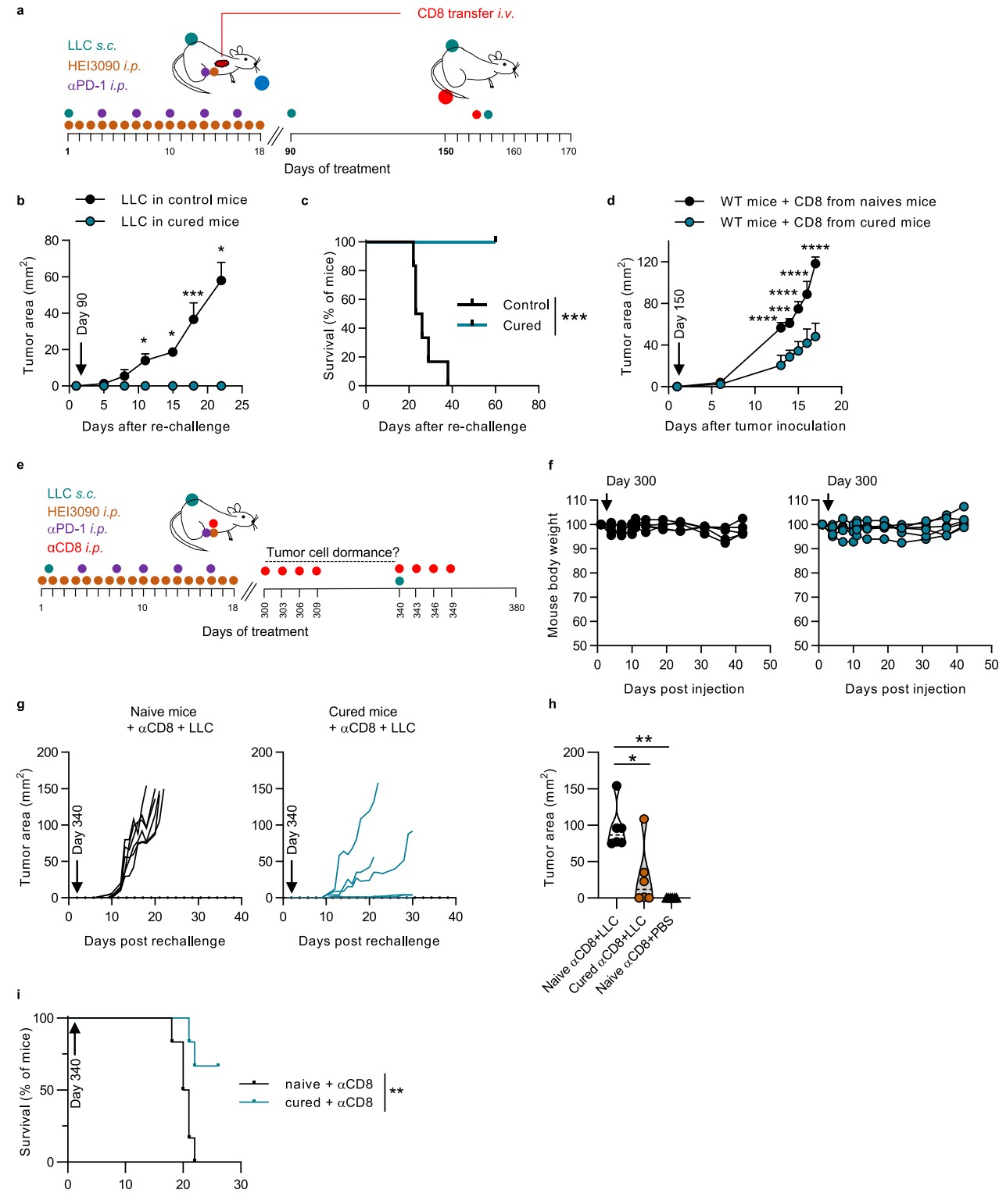

samples[26] and showed that P2RX7 is functional in leukocytes whereas it is nonfunctional in tumor cells. Considering that P2RX7 is a pro-apoptotic receptor, it makes sense that tumor cells express a nonfunctional receptor. Whether this nonfunctional receptor corresponds to the non-conformational P2RX7 (nfP2RX7), described to be expressed by tumor cells[27], remains to be determined as well as the effect of HEI3090 on nfP2RX7.

Despite the finding that P2RX7 expression by immune cells restrains tumor growth[9,10], the use of specific P2RX7 antagonists

has been promoted to treat cancers on the basis that inhibition of tumor cell proliferation would be more efficient[28,29]. Considerable effort has been made to engineer-specific P2RX7 antagonists[30] and two of them (A74003 and AZ10606120) inhibited B16 tumor growth in immunocompetent mice[10]. However, to our knowledge, these compounds have not been tested to treat cancer and have failed in the first clinical trials to treat inflammatory and pain-related diseases[30]. In addition, in the preclinical mouse model, we were unable to inhibit LLC and B16-F10 tumor growth

**Fig. 6 HEI3090 combined with αPD-1 induces antitumor memory immune response. a** Schematic illustration of treatments with transfer of CD8 cells. **b** Average tumor area of LLC allograft in 90-day-old WT and 90-day-old cured mice in absence of treatment. Curves showed mean tumor area in mm² ± SEM ($n = 7$ per group, two-way Anova test). **c** Survival curves of animals from the study shown in **b**. Curves showed survival ($n = 7$ per group, Mantel Cox test). **d** Average tumor area of LLC allograft in WT mice injected with CD8⁺ T cells isolated from rechallenged cured mice as shown in **a**. Curves showed mean tumor area in mm² ± SEM ($n = 4$ per group, two-way Anova test). **e** Long-lasting antitumor immune response: schematic illustration of treatments. **f** Mouse body weight follow up of 300-day-old cured mice injected with anti-isotype (black circle) or depleting αCD8 antibodies (blue circle). Each curve represents one mouse ($n = 6$ per group). **g** Individual survival curves of 340-day-old WT and cured animals injected with anti-CD8 antibody and rechallenge with LLC in absence of treatment. ($n = 6$ per group). **h** Average tumor area from animals shown in **g**. Data are presented by violin plots showing all points with hatched bar corresponding to median of tumor area ($n = 6$ per group. Two-tailed Mann–Whitney test). **i** Survival curves from animals shown in **g**. ($n = 6$ per group, Mantel Cox test). p values: *$p < 0.05$, **$p < 0.01$ ***$p < 0.001$, ****$p < 0.0001$. Source data are provided as a Source Data file.

when we tested the GSK1370319A compound, a well-characterized P2RX7 antagonist[19]. In line with this finding, our present results suggest that facilitation of P2XR7 is associated with efficient antitumor immunity in two different models of transplantable tumor (expressing moderate or higher level of P2RX7) as well as in the LSL-KRas^G12D genetic lung cancer mouse model. These results illustrate the view that P2RX7 activation, rather than inhibition, represents a promising strategy in cancer immunotherapy to unleash the immune responses, notably in conjunction with anti-checkpoint blockade. Therapeutic antibody represents another promising field of investigation to treat cancer. In particular, Gilbert et al. described an antibody against a nonfunctional P2RX7 variant that is promising to treat basal cell carcinoma[31]. It would be interesting to combine HEI3090 with the therapeutic P2RX7 antibody and assay the efficacy of this combo treatment.

Antineoplastic action of eATP was previously explored using ATP administration in cancer patients and abandoned for lack of convincing results[32,33]. Extracellular ATP is naturally degraded to adenosine by ectoenzymes, and adenosine is an immunosuppressive molecule[34]. To inhibit the production of adenosine, blocking antibodies against CD39 and CD73 ectoenzymes were produced and tested in mouse cancer models but also in ongoing clinical trials (NCT03454451). This strategy seems to be promising, at least, in mice tumor models. In a first study, Perrot et al. showed that antibodies targeting human CD39 and CD73 promoted antitumor immunity by stimulating DC and macrophages which, in turn, restored the activation of effector T cells[35]. The authors also reported that the combination of anti-CD39 monoclonal antibody with oxaliplatin increased the survival of tumor bearing mice, at least for 50 days. In a second study, an independent anti-CD39 antibody was generated and tested on different mouse tumor models. This antibody alone dampened tumor growth and when combined with αPD-1, it further slowed tumor progression and 50% of the mice showed a complete rejection[36]. Mechanistically, the anti-CD39 antibody treatment led to an increased eATP levels via the P2RX7/NLRP3/IL-18 to stimulate myeloid cells. Next, the authors demonstrated that anti-CD39 antibody sensitized αPD-1 resistant tumors by increasing CD8⁺ T cells infiltration. Our results confirm these findings but also bring additional highlights. First, we showed that activation of the eATP/P2RX7/NLRP3/IL-18 pathway by HEI3090 increased long-lasting immune responses when combined with αPD-1 antibody. Second, we demonstrated that the endogenous eATP levels present in the TME were sufficient to enhance P2RX7's activation in the presence of HEI3090. These conditions are ideal to allow P2RX7 activation where it is needed and avoid the possible adverse effects associated with a systemic increase of ATP levels, such as the one observed in response to anti-CD39 and -CD73 antibodies.

It was shown that eATP attracts DC precursors toward the TME and promotes their activation state and their capacity to present antigen[37,38]. During this study, we showed that HEI3090 targets P2RX7-expressing immune cells, especially phagocytic cells, such as macrophages and DCs (Supplementary Fig. 6a). Between macrophages and DCs, DCs were the most promising candidate; they express high levels of P2RX7, they are able to release IL-18, and they are professional antigen-presenting cells able to induce a potent antitumor immune response. We therefore tested their involvement by doing an adoptive transfer of WT DC in p2rx7⁻/⁻ mice. Doing so, we restored responsiveness to HEI3090 (Fig. 3b). We also observed that cDC CD4⁺ from mice treated with HEI3090 expressed higher levels of P2RX7 (Supplementary Fig. 4c). Collectively, these results demonstrated that DCs mediate HEI3090's antitumor activity, but macrophages may have a secondary role in this effect.

Intriguingly, we did not observe an enhanced production of mature IL-1β in mice treated with HEI3090 (Fig. 4d). This was unexpected as secretion of mature IL-1β depends on the ATP/P2RX7-induced NLRP3 inflammasome activation as well[39]. However, unlike IL-1β, the inactive precursor form of IL-18 is constitutively expressed in most human and animal cells. Whether this explanation is sufficient to account for this differential IL-1β/IL-18 production is currently not known.

Whereas IL-1β is described to induce immune escape[40], IL-18 is involved in Th1 polarization and NK cell activation. We showed here that IL-18 produced in response to HEI3090 treatment orchestrated the antitumor immune response by driving IFN-γ production by NK and CD4⁺ T cells. This is in line with the well-known IFN-γ stimulating activity of IL-18 (originally designated as IFN-γ-inducing factor), and with its Th1 and NK cells stimulating activity[41,42]. Protective effect of IL-18, but also the activation of NLRP3, have been previously reported in various mouse cancer models[43,44] NLRP3 activation in DCs as well as IL-18 have been linked to better prognosis, to drive antitumor immunity and to enhance the efficacy of immunotherapies in different tumor models[45,46]. In fact, when we combined HEI3090 with an αPD-1 antibody, we observed that the combo therapy efficiently controlled tumor burden in the three cancer models studied. Notably, the combo treatment cured 80% of LLC tumor bearing mice and very interestingly, cured mice developed an antitumor memory response.

CD8 memory T cells, comprising the circulating memory pool —composed of effector memory (T_EM) and central memory (T_CM) cells—and the tissue resident (T_RM) pool, play crucial roles in antitumor memory responses[47]. We showed that circulating CD8⁺ T cells participated in cancer immunosurveillance after HEI3090 treatment (Fig. 6d). However, this CD8⁺ T cells pool cannot be responsible for the entire response, since antitumor responses were still effective when CD8⁺ T cells were depleted (Fig. 6g). These results suggest that other immune cells participate in local cancer surveillance. Possible candidates are the non-recirculating CD8⁺ T_RM cells. The persistence of T_RM cells in tissues has been shown to depend on signaling programs driven by TGFβ and Notch-dependent signaling signature. Whether HEI3090 directly stimulates those programs remains to be

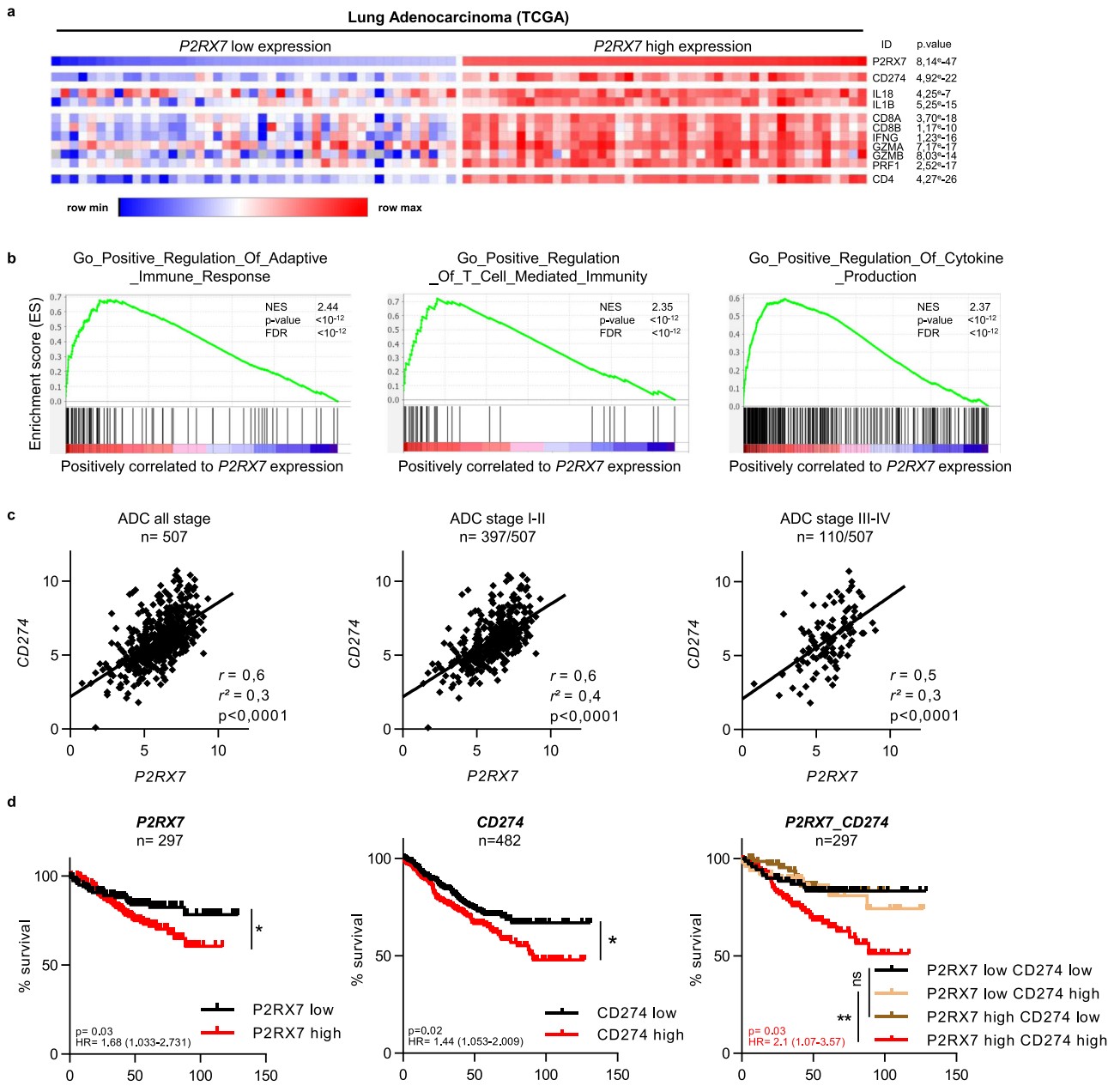

**Fig. 7 P2RX7 expression in LUAD is associated with "hot" immunophenotype signature. a** Association of *P2RX7* mRNA expression with a cluster of inflammatory genes (heatmap). Expression values are represented as colors, where the range of colors (red, pink, light blue, dark blue) shows the range of expression values (high, moderate, low, lowest). Raw *p* values (Linear models for microarray analysis, Limma) are shown. **b** Gene set enrichment analysis (GSEA) plot associating *P2RX7* high mRNA levels from LUAD patients (TCGA) with three inflammatory signatures. The enrichment score is shown as a green line, and the vertical black bars below the plot indicate the positions of specific inflammatory signature-associated genes, which are mostly grouped in the fraction of upregulated genes. For each signature, normalized enriched score (NES), *p* values (bilateral Kolmogorov–Smirnov), and false discovery rate (FDR) are shown. **c**. Correlation curves of *P2RX7* and *CD274* expression from LUAD patients (TCGA) of all stage (left panel), low stage (middle panel), and high stage (right panel). *r*, *r*², and *p* values are shown in each panel, (Person correlation and *t* test). **d** Kaplan–Meyer plot (http://kmplot.com) showing survival curves of *P2RX7* high vs. *P2RX7* low patients (left panel), *CD274* high vs. *CD274* low (middle panel), and *P2RX7* high or low vs. *CD274* high or low (right panel). For all panels, the optimal cutoff is determined on KMplot. The *p* value (log-rank, Mantel Cox test), the hazard ratio, and number of patients are indicated. Source data are provided as a Source Data file. ADC adenocarcinoma, HR hazard ratio.

determined but we observed, using HEK mP2RX7 cells, that HEI3090 enhanced ATP-stimulated ERK pathways. Our results are also compatible with a role for CD4⁺ T memory cells and the setup of a humoral response, in which B lymphocytes produce antibody against tumor cells.

Therapy with different αPD-1/PD-L1 antibodies was approved in NSCLC in the first- and second-line settings. However, a significant fraction of patients does not benefit from the treatment (primary resistance), and some responders relapse after a period of response (acquired resistance)[48]. Expression of PD-L1 per se is not a robust biomarker with a predictive value since the αPD-L1 response has also been observed in some patients with PD-L1-negative tumors. Improvement of patient management for immunotherapy undoubtedly relies on the identification of such

predictive markers. Using TCGA dataset, we uncovered that *P2RX7* expression is correlated to *CD274* (PD-L1) expression and "hot" immunophenotype signatures in NSCLC patients. In addition, patients with high *P2RX7* and low *CD274* or high *CD274* and *low P2RX7* have a better overall survival than patients with high *CD274* and high *P2RX7*. This result suggests that immunotherapies may be efficient in double positive patients and questions the ability of P2RX7 to represent a valuable biomarker for αPD-1/PD-L1 therapies. In this context, we showed in another study[26] that the expression of *P2RX7B* splice variant in tumor immune cells is associated with less infiltrated tumors in lung adenocarcinoma. Mechanistically, we observed that the differential expression of the *P2RX7B* splice variant in immune cells within tumor area correlates with the expression of a less functional P2RX7 and lower leukocytes recruitment into LUAD.

We demonstrated that a small-molecule activator of P2RX7 boosts immune surveillance by unleashing the effector functions of adaptive immune T cells and improving the efficacy of αPD-1 treatment. This therapeutic strategy holds new hopes for cancer patients; by increasing tumor immunogenicity, it could first increase the number of patients eligible to immunotherapies and second, it could also be used as a neoadjuvant or adjuvant therapies of locally advanced lung tumors.

## Methods

**Mice**. Mice were housed under standardized light–dark cycles in a temperature-controlled air-conditioned environment under specific pathogen-free conditions at IRCAN, Nice, France, with free access to food and water. All mouse studies were approved by the committee for Research and Ethics of the local authorities (CIEPAL #28, protocol numbers MESRI 23707, 13656) and followed the European directive 2010/63/UE, in agreement with the ARRIVE guidelines. Experiments were performed in accord with animal protection representative at IRCAN. *P2rx7−/−* (B6.129P2-*P2rx7*tm1Gab/J) and *il18−/−* mice were from the Jackson Laboratory. LSL-*KRas*G12D are from the Jackson Laboratories (ref 008179). *P2rx7-flox* mice were engineered as follow: ES clones (C57/BL/6) containing a construct for the conditional elimination or re-expression of P2RX7 (purchased from The European Conditional Mouse Mutagenesis Program) were injected into blastocytes of C57Bl6/N, chimeric mice were selected and crossed with deleter mice that are transgenic for the Flip-recombinase under the control of the ubiquitous *Actin* promoter to produce *p2rx7*loxP/loxP mice. Our *p2rx7*loxP/loxP mice, with loxP sequence floxing the second exon of *p2rx7*, were crossed with (C57BL/6NTacGt (ROSA)26Sor<tm1(ACTB-Cre,-EGFP)) transgenic mice which express the Cre recombinase under the control of the b-actin promoter to produce *p2rx7*exon2−/− mice or with *LysM-Cre* mice (B6.129P2-Lys2tm1(cre)lfo from the Jackson Laboratories (obtained from Dr B. Chazaud, France) to generate myeloid cell conditional *p2rx7* knockout and WT control (*p2rx7*loxp/loxp) mice. Control C57BL/6J OlaHsD female (WT mouse) was supplied from Envigo (Gannat, France).

**In vivo treatments**. Five $10^5$ tumor cells were injected s.c. into the left flank of WT mice. Pharmacokinetic analysis (Fig. 1g), to characterize the clearance of HEI3090 showed that after a period of 18-h HEI3090 concentration is <10 nM. Therefore, we have decided to inject HEI3090 daily. Mice were treated i.p. with vehicle (PBS, 10% DMSO) or with HEI3090 (1.5 mg/kg in PBS, 10% DMSO), which corresponds to the highest soluble dose. For therapeutic settings, treatment started at day 3, when tumor reached ~10–15 mm$^2$, for a maximum of 20 days and mice received vehicle or HEI3090 (3 mg/kg in PBS, 10% DMSO) daily. Depleting and neutralizing antibodies from BioXCell were given i.p. in the right flank at days −1, 3, 7, and 10. αPD-1 antibody was given i.p. at days 4, 7, 10, and 13 (or as stated in the legend of the figure) post-tumor cell inoculation. Antibodies are listed in Supplementary Table 1. αPD-1 and HEI3090 were injected separately, with at least 30 min delay between the two injections. Two hundred microliters liposome clo-dronate (Liposoma) were injected i.p. 3 days before LLC tumor cell inoculation in WT mice and then every 3 days, at least 1 h before HEI3090 treatment after the treatment started. CD8$^+$ T cells were sorted from peripheral lymph nodes of cured or naïve WT mice with Dynabeads® Untouched™ Mouse CD8 Cells (Invitrogen) according to the supplier's instructions. $5.10^5$ CD8$^+$ T cells were adoptively transferred into 8-week-old naïve WT mice (i.v.) 1 day before tumor inoculation. $5.10^5$ LLC cells were injected s.c. into the left flank of these mice and given no further treatment.

Intratracheal delivery of adenoCre induces oncogenic KRAS in lung airway cells, leading to multifocal adenocarcinomas and a median survival of about 6 months[49]. Starting with tumors established for 3 months in adult LSL-*Kras*G12D mice of either gender, treatment with vehicle or 1.5 mg/kg HEI3090 daily by i.p. injection was performed for 21 additional days.

**Adoptive transfer in *p2rx7*-deficient mice**. Spleens from WT C57BL/6J female mice were collected and digested in RPMI 1640 medium containing 5% FCS, 1-mg/ml collagenase IV (Sigma-Aldrich), and 50 U/ml DNase I (Roche) for 7 min at 37 °C. Single-cell suspensions of spleens were prepared by passage through 100 μm cell strainers (BD Biosciences) and counted. For WT DCs isolation, spleens were digested with the spleen dissociation kit (Miltenyi Biotech) and isolated with the CD11c Microbeads UltraPure (Miltenyi biotech) according to the supplier's instructions. $5.10^6$ splenocytes or $1.2.10^6$ DCs were injected i.v. in *p2rx7−/−* mice 1 day before subcutaneous injection of $5.10^5$ LLC cells into the left flank. Mice were treated i.p. every day for 12 days with vehicle (PBS, 10% DMSO) or with HEI3090 (1.5 mg/kg in PBS, 10% DMSO). At day 12, tumors were collected, weighted, and digested, when flow cytometry analyses were done.

**Flow cytometry and antibodies**. Tumors were mechanically dissociated and digested with 1-mg mL$^{-1}$ collagenase A and 0.1-mg mL$^{-1}$ DNase I for 20 min at 37 °C. Then single-cell suspensions of tumors were prepared by passage through 100 μM cell strainers (BD Biosciences). Surface staining was performed by incubating cells on ice, for 20 min, with saturating concentrations of labeled Abs in PBS, 5% FCS, and 0.5% EDTA. After blocking Fc receptors using anti-CD16/32 antibodies, cells were stained with the appropriate combination of antibodies (see Supplementary Table 1). The transcription factor staining Buffer Set (eBioscience) was used for the FoxP3 staining. For intracellular cytokines, staining was performed after stimulation of single-cell suspensions with Phorbol 12-myristate 13-acetate (PMA at 50 ng mL$^{-1}$, Sigma), ionomycin (0.5 μg mL$^{-1}$, Sigma) and 1 μL mL$^{-1}$ Golgi Plug™ (BD Biosciences) for 4 h at 37 °C 5% CO$_2$. Cells were incubated with Live/Dead Near-IR stain (Invitrogen), according to the manufacturer's protocol prior to Ab surface staining. Then, intracellular staining was performed using Cytofix/Cytoperm™ kit (BD biosciences) following the manufacturer's instructions. The production of IFN-γ and IL-10 was simultaneously analyzed in CD45$^+$, NK, CD4$^+$ T, or CD8$^+$ T cells. Data files were acquired and analyzed on Aria III using Diva software (BD Biosciences) or on the CytoFlex LX (Beckman Coulter) and analyzed using FlowJo software (LLC). Gating strategies are shown in Supplementary Fig. 12.

**Immunohistological analysis of tumors**. Collected tumors or lungs were fixed in 3% formamide for 16 h prior inclusion in paraffin. We used the following antibodies: anti-CD3, anti-CD8, anti-IL-18, αPD-L1, and anti-Ki67 (see Supplementary Table 1). After staining, slides were captured and analyzed using NDP view2-software. For the analyses, five zones per tumor were randomly selected and cells were counted using ImageJ software. Results are expressed as number of positive cells per total cell number.

**Characterization of lung lesions in the LSL-*KRas*G12D mouse model**. At the end of the treatment mice were sacrificed, exsanguinated and lungs processed for histologic and immunological analyses. After deparaffinization, HE stains were performed and slides were captured and analyzed using NDP view2 software. Tumor burden was calculated by determining the mean of total tumor area per lung using the NDP view2 software. To count the cells and determine the percentage of Ki67-positive cells within lesions, ten lesions per lung, from grade 2 to 5 according to the Sutherland scoring[50], were randomly selected, their perimeter was determined, and positive and negative nuclei were counted using ImageJ software. Results are expressed as number of cells per mm$^2$ and the percentage of Ki67-positive cells.

**Ex vivo macrophages and BMDC stimulation**. Peritoneal lavage was done with RPMI 1640 medium on WT or *p2rx7−/−* mice. $4.10^5$ macrophages were seeded in a 96-well plate overnight in RPMI 1640 containing 10% FBS, 2% sodium pyruvate, 1% penicillin/streptomycin, and 50-μM β-mercaptoethanol. After two washes with the complete medium, cells were primed for 4 h with 100 ng/ml LPS (Sigma-Aldrich) at 37 °C and then stimulated for 30 min at 37 °C with ATP (Sigma-Aldrich) with or without 50 μM of HEI3090 or with 10 μM nigericin. When indicated, NLRP3 inflammasome was inhibited with 1 μM of MCC950 (Invivogen) for 1 h at 37 °C before cell stimulation. To prepare BMDC, leg bones were removed from C57Bl/6 mice, cut with scissors, and flushed with sterile PBS pH 7.4 via syringe. Bone-marrow-derived dendritic cells (BMDCs) were obtained from bone-marrow cells seeded in Petri dishes and cultured in RPMI medium containing 10% fetal calf serum, 2 mM L-Glutamine, 50-U/mL Penicillin, 50 μg/mL Streptomycin and 20% conditioned medium from GM-CSF-producing J558L cells. Medium was refreshed every 3 days. On the 7th day of the differentiation protocol, semi-adherent BMDCs were collected using PBS containing 10 mM EDTA and re-plated in new Petri dishes with fresh medium. Mature and semi-adherent BMDCs were used for experiments on the 14th day of culture. Supernatants were collected and stored at −80 °C before cytokine detection by ELISA using mouse IL-1 beta/IL-1F2 (R&D) and IL-18 (MBL) according to the supplier's instructions.

Cells were lysed with Laemmli buffer (10% glycerol, 3% SDS, 10 mM Na$_2$HPO$_4$) with protease inhibitor cocktail (Roche). Proteins were separated on a 12% SDS-PAGE gel and electro transferred onto PVDF membranes, which were blocked for 30 min at RT with 3% bovine serum albumin. Membranes were incubated with primary antibodies diluted 1/1000 at 4 °C overnight. The following antibodies were

used: anti-NLRP3, anti-ASC, anti-caspase-1, and anti-β-actin (see Supplementary Table 1). Secondary antibodies (Sigma-Aldrich) were incubated for 1 h at RT. Immunoblot detection was achieved by exposure with a chemiluminescence imaging system (PXI Syngene, Ozyme) after membrane incubation with ECL (Immobilon Western, Millopore). The bands intensity values were normalized to that of β-actin using ImageJ software. Full scan blots are presented in the Source Data file.

**Cell lines**. The LLC cell line (ATCC CRL-1642), the melanoma B16-F10 (ATCC CRL-6475) cell line, and HEK293T cells (ATCC CRL-3216) were used in this study. Cells were cultured in DMEM medium supplemented with 10% FBS and 100 U/mL penicillin and 100 mg/mL streptomycin at 37 °C in a humid atmosphere containing 5% $CO_2$ and routinely checked for mycoplasma contamination. Cells were used between passages 5 and 15.

**Calcium uptake assay**. $20.10^3$ HEK293T-mP2RX7$^{C57BL/6J}$ or HEK293T-pcDNA6 cells were seeded per well on a poly-L-Lysine (Sigma-Aldrich) 96 well-coated plate in complete medium. Twenty-four hours later, cells were washed in sucrose buffer (300 mM sucrose, 5 mM KCl, 1 mM $MgCl_2$, 1 mM $CaCl_2$, 10 mM Glucose, 20 mM HEPES, pH 7-7.4) and incubated for 1 h at 37 °C with 1 μM of Fluo-4 AM (Life Technologies). Cells were washed once with PBS + 5% FBS then twice with the sucrose buffer. Fluo-4 AM fluorescence was read on a Xenius, microplate reader (SAFAS) at 485/528 nm at 37 °C. After 3 min of baseline readings, 333 μM of ATP (Sigma-Aldrich) were added with or without various concentrations of HEI3090. For the assay on splenocytes, spleens were digested with the spleen dissociation kit (Miltenyi Biotech). $5.10^5$ splenocytes were seeded per well on a 96-well plate in sucrose buffer and incubated for 30 min at RT with 1 μM of Fluo-4-AM.

**TO-PRO-3 uptake assay**. $30.10^3$ HEK293T-mP2RX7$^{C57BL/6J}$ or HEK293T-pcDNA6 cells were seeded per well on a poly-L-Lysine (Sigma-Aldrich) black clear bottom 96 well-coated plate (Perkin Elmer) in complete medium. Twenty-four hours later, cells were washed twice in sucrose buffer (300 mM sucrose, 5 mM KCl, 1 mM $MgCl_2$, 1 mM $CaCl_2$, 10 mM Glucose, 20 mM HEPES, pH 7-7.4). One microliter of TO-PRO-3 (Life Technologies) was added in the sucrose buffer. TO-PRO-3 fluorescence was read on a Xenius, microplate reader (SAFAS) at 550/660 nm at 37 °C. After 10 min of baseline readings, 250 μM of ATP (Sigma-Aldrich) were added with or without various concentrations of HEI3090. Alternatively, percentage of TO-PRO-3$^+$ cells was analyzed on non-adherent cells by flow cytometry.

**Cell viability**. Colorimetric assay based on XTT (Roche) was used to quantify the viability of tumor cells treated with 1 mM of BzATP and 50 μM of HEI3090. LLC cells were treated for 16 h and B16-F10 for 3 h. Cellular viability was determined as described in the supplier's protocol.

**In vivo assay for ICD**. B16-F10 was exposed to 3 mM ATP and 50 μM HEI3090 for 3 h at 37 °C. Cells were then washed and resuspended in PBS, and cell death was determined with trypan blue. Dying B16-F10 cells reached 97% in this assay. $1.10^5$ dying cells were injected s.c. into the right flank of WT mice (in 200-μL PBS). Control mice received 200 μL PBS into the right flank. Seven days later, mice were challenged with live $5.10^5$ B16-F10 cells into the left flank. Tumor growth was routinely monitored at both injection sites.

**Synthesis of HEI3090**. Starting materials are commercially available and were used without further purification (suppliers: Carlo Erba Reagents S.A.S., Thermo Fisher Scientific Inc., and Sigma-Aldrich Co.). Intermediates were synthesized[28] and melting points were measured on the MPA 100 OptiMelt® apparatus and are uncorrected. Nuclear magnetic resonance (NMR) spectra were acquired at 400 MHz for $^1$H NMR and at 100 MHz for $^{13}$C NMR, on a Varian 400-MR spectrometer with tetramethylsilane (TMS) as internal standard, at 25 °C. Chemical shifts (δ) are expressed in ppm relative to TMS. Splitting patterns are designed: s, singlet; d, doublet; dd, doublet of doublet; t, triplet; m, multiplet; sym m, symmetric multiplet; br s, broaden singlet; br t, broaden triplet. Coupling constants (J) are reported in Hertz (Hz). Thin layer chromatography (TLC) was realized on Macherey Nagel silica gel plates with fluorescent indicator and were visualized under a UV lamp at 254 and 365 nm. Column chromatography was performed with a CombiFlash Rf Companion (Teledyne-Isco System) using RediSep packed columns. IR spectra were recorded on a Varian 640-IR FT-IR Spectrometer. Elemental analyses (C, H, N) of new compounds were determined on a Thermo Electron apparatus by "Pôle Chimie Moléculaire-Welience," Faculté de Sciences Mirande, Université de Bourgogne, Dijon, France. LC-MS was accomplished using an HPLC combined with a Surveyor MSQ (Thermo Electron) equipped with APCI source.

The synthesis of the title compound was accomplished starting from L-pyroglutamic acid also known as the "forgotten amino acid," bio-sourced affordable raw material (Fig. 1a). After simple esterification of the L-pyroglutamic acid, the resulting methyl pyroglutamate was reacted with 2,4-dichlorobenzylamine in presence of catalytic amount or zirconium (IV) chloride in solvent-less

conditions to provide pyroglutamide (HEI2313) in 90% yield. To obtain (S)-N$^1$-(6-chloropyridin-3-yl)-N$^2$-(2,4-dichlorobenzyl)-5-oxopyrrolidine-1,2-dicarboxamide (HEI3090), a mixture of pyroglutamide (HEI2313) (1.86 g, 6.48 mmol) and 2-chloro-5-isocyanatopyridine (1.00 g, 6.48 mmol) in toluene was refluxed for 24 h under nitrogen atmosphere and magnetic stirring. After cooling to room temperature, the mixture has been concentrated in vacuo and the resulting crude has been purified by column chromatography ($CH_2Cl_2$/MeOH: 1/0 to 9/1) to afford pure HEI3090 as a white powder in 54% yield (1.62 g, 3.51 mmol). mp 187–190 °C (MeOH); TLC Rf ($CH_2Cl_2$/MeOH: 95/5) 0.8; $^1$H NMR (CDCl$_3$, 400 MHz) δ ppm 2.21–2.37 (m, 2H, $CH_2CH_2CH$), 2.59–2.68 (m, 1H, $CH_2CH_2CH$), 2.99–3.10 (m, 1H, $CH_2CH_2CH$), 4.49 (dd, J = 15.2, 6.2 Hz, 1H $NHCH_2$), 4.55 (dd, J = 15.2, 6.2 Hz, 1H, $NHCH_2$), 4.77 (dd, J = 7.2, 2.8 Hz, 1H, $CH_2CH_2CH$), 6.65 (br t, J = 6.2 Hz, 1H, $NHCH_2$), 7.22 (dd, J = 8.1, 2.0 Hz, 1H, ArH), 7.29 (d, J = 8.6 Hz, 1H, ArH), 7.33 (d, J = 8.1 Hz, 1H, ArH), 7.38 (d, J = 2.0 Hz, 1H, ArH), 7.92 (dd, J = 8.6, 2.5 Hz, 1H, ArH), 8.49 (d, J = 2.5 Hz, 1H, ArH), 10.66 (br s, 1H, NHAr); $^{13}$C NMR (CDCl$_3$,100 MHz) δ ppm 21.4 ($CH_2$), 32.4 ($CH_2$), 41.4 ($CH_2$), 59.2 (CH), 124.3 (CH), 127.5 (CH), 129.5 (CH), 130.2 (CH), 130.9 (CH), 133.1 (C), 133.6 (2C), 134.2 (C), 141.3 (CH), 146.2 (C), 150.3 (C), 170.0 (C), 177.7 (C); IR (neat): ν cm$^{-1}$ 3271, 3095, 1720, 1655, 1594, 1542, 1464, 1217, 1104, 829.

**Statistical analyses**. All analyses were carried out using Prism software (Graph-Pad). Mouse experiments were performed on at least $n = 5$ individuals, as indicated in Fig legends. Mice were equally divided for treatments and controls. Data were represented as mean values and error bars represent SD or SEM. Mann–Whitney, t-test, and Mantel Cox were used to evaluate the statistical significance between groups. The corresponding two-way Anova tests and p values were mentioned in the legend of each figure. For survival analysis, patients were separated based on optimal cutoff of the expression value of the marker determined using KMplot.

## Data availability

In silico data used in this study are available in the Tumor Comprehensive Genome Atlas (TCGA) project database, at https://software.broadinstitute.org/morpheus/. We used the TCGA Lung Adenocarcinoma (LUAD) cohort.The Gene Set Enrichment Analysis (GSEA) signatures sets are available at https://www.gsea-msigdb.org/gsea/msigdb/index.jsp and into the folder named « GSEA_SIGNATURE_SET » as source data files named:« GO_POSITIVE_REGULATION_OF_ADAPTIVE_IMMUNE_RESPONSE »« GO_POSITIVE_REGULATION_OF_CYTOKINE_PRODUCTION »« GO_POSITIVE_REGULATION_OF_T_CELL_MEDIATED_IMMUNITY »« GO_RESPONSE_TO_TUMOR_CELL »The GSEA analyses results are available into the folder named « GSEA_ANALYSES » as Excel Data Sheets named:« ANALYSIS_GO_POSITIVE_REGULATION_OF_ADAPTIVE_IMMUNE_RESPONSE »« ANALYSIS_GO_POSITIVE_REGULATION_OF_CYTOKINE_PRODUCTION »« ANALYSIS_GO_POSITIVE_REGULATION_OF_T_CELL_MEDIATED_IMMUNITY »« ANALYSIS_GO_RESPONSE_TO_TUMOR_CELL »Source data are available as a Source Data file. The remaining data are available within the Article, Supplementary Information or available from the authors upon request. Source data are provided with this paper.

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

## Acknowledgements
The authors wish to thank Prof L. Counillon for valuable discussions, Anne Laure Rossi for technical expertise and graphical art, and Alexandre Gallerand for BMDC preparation. The authors greatly acknowledge the IRCAN's Animal core facility and IRCAN's Flow Cytometry Facility that is supported by FEDER, Ministère de l'Enseignement Supérieur, Région Provence Alpes-Côte d'Azur, Conseil Départemental 06, ITMO Cancer Aviesan (plan cancer), Canceropole PACA, CNRS, and Inserm.The funding sources for this work were Institut National du Cancer (INCa, plan Cancer), Canceropole PACA, Bristol-Myers Squibb Foundation for Research in Immuno-Oncology, the Ligue Nationale Contre le Cancer, the French Government (National Research Agency, ANR) through the "Investments for the Future" programs LABEX SIGNALIFE ANR-11-LABX-0028 and IDEX UCAJedi ANR-15-IDEX-01 and the Centre National de la Recherche Scientifique (CNRS), the Institut National de la Santé et Recherche Médicale (INSERM) and the University of Orleans, The Region Centre Val de Loire (2003-00085470), the Conseil General du Loiret and European Regional Development Fund (FEDER No. 2016-00110366 and EX005756).

## Author contributions
L.D., S.J.H., J.B., L.S., X.D., and V.V.-C. conceived and design the study. L.D., S.J.H., X.D., C.Du., J.K., C.D., N.R., C.F., G.H., A.G., and V.V.-C. developed the methodology. L.D., S.J.H., J.B., L.S., T.J., X.D., B.R., J.K., C.Du., A.G., and J.C.-V. acquired the data (provided animals, provided facilities, and so on). L.D., S.J.H., J.B., L.S., X.D., N.R., C.F., G.H., J.C.-V., R.M., S.A., A.G., P.H., and V.V.-C. analyzed and interpreted the data. S.J.H. and V.V.-C. wrote the manuscript. All authors reviewed the manuscript.

## Competing interests
The authors declare no competing interests.
