## [Peer Review File · Nature Communications]

Reviewers' comments:

Reviewer #1 (Remarks to the Author): expertise in purinergic receptors, immune cells, cancer, immunotherapy

The authors report on a novel pharmacological modulator of P2RX7 receptor that potentiates anti-PD-1 treatment to effectively control the growth of lung tumors in an oncogene-induced NSCL and a transplantable NSCLC model leading to long lasting antitumor immune responses. The authors show that the mechanism is based on IL-18 production via P2X7 activation which leads to the production of IFN γ by Natural Killer and CD4+ T cells within the TME. The study is interesting and has clinical potential.

1. P2X7 activation also triggers IL-1beta production and IL-1beta was shown to cause immune escape in different solid tumors such as breast cancer (Kaplanov I et al. PNAS 2019). Did the authors study IL-1beta production?
2. P2X7 activation leads to INflammasome activation -was Nlrp3 activation analyzed, e.g. caspase-1 cleavage?
3. Figure 4F: it is questionable to quantify a cytokine by IHC. During the tissue processing most cytokines are washed out.
4. What cell type produces IL-18 in the TME?
5. Figure 5D: it is hard to believe that blocking IL-18 completely blocks the P2X7 activation induces effects on IFN γ pos cells. P2X7 activation causes Nlrp3 activation and IL1beta release which is not blocked by anti IL18.
6. Figure 6I: the follow-up of 3 weeks is too short.
7. IN the gene expression study, Figure 7S you also find IL-1beta upregulated in the P2X7 group - why should that be functionally irrelevant?

Reviewer #2 (Remarks to the Author): expertise in lung cancer, immunotherapy, TME

This manuscript examines the effects of a potential novel inhibitor of P2RX7 as a potential cancer immunotherapy. While the studies are of potential interest, there are a number of problems with the studies that need to be addressed. In particular for many experiments there is a lack of experimental detail, such as dose of drug used or the time point examined. Specific points are addressed below.

Major points:

1. There is a significant literature suggesting that P2RX7 inhibitors may represent a therapeutic approach in cancer (Young et al Frontiers Chem 2018 for a review). Thus there is the possibility of opposing effects of activating this channel in cancer cells vs the TME. The data in this paper have focused on the stromal effects, but there at least needs to be some discussion of the literature supporting the use of inhibitors of P2RX7 vs activators. In that regard it would be important to know if the cancer cells used in this study express these channels.
2. In Fig. 1 the authors have used a cell line that is engineered to overexpress P2RX7 as a screen for activators. It is likely that the levels of expression, which are not shown are supraphysiologic. Therefore it would be of interest to determine the effects of this agent in a cell line that normally expresses P2RX7. In addition, there is little indication as to the specificity of this agent. Does it regulate any other ATP regulated ion channels? The authors present data indicating formation of a

large pore in the setting of eATP and HEI3090

(Fig. 1D). However, there is no indication what the three tracings represent. Furthermore, the data shown in Fig. 1E demonstrate a much more modest effect on large pore formation.

3. Since much of the studies presented are *in vivo*, it is important to determine the dose that can be achieved. This is shown in Fig. 1F, where the maximum effective dose is 2 micromolar.

However, there are no details provided as to what the administered dose was, and there are concerns regarding the relative transient nature of achieving this dose. More details need to be provided.

4. In Fig. 2 the authors examine the effects of HEI3090 in a KRas mouse model of lung cancer. Data shown in Pnaels E-H suggest additivity or synergy with anti-PD-1 therapy. However, there are no untreated animals. This is a particular concern, since several reports have indicate that these GEMM models are unresponsive to anti-PD-1 (McFadden et al PNAS 2016). Thus it is not clear from these data if we are just seeing a modest inhibitory effect, or additivity with anti-PD-1.

5. The characterization of T cells in the tumors shown in Fig 3 is incomplete. It would be important to see the gating strategy and to quantify both CD8 and CD4 populations.

6. The authors present extensive data demonstrating that the effects of activating P2RX7 are mediated through IL18, and do not have a role for IL1beta in this pathway. In fact, no increases in IL1beta are detected. If activation of the inflammasome and caspase1 are medatied by P2RX7, one would anticipate at least seeing levels of ILbeta increase. This needs to be discussed.

7. The TCGA data shown in Fig. 7 is difficult to interpret, due to the fact that these data cannot identify the cell type on which P2RX7 is expressed. The observation of poorer survival in patients with high P2RX7 expression might suggest that activators of this channel would not be an attractive therapeutic strategy

8. The models used in this study indicate a role for activation of P2RX7 in immune cells. However, this may be model dependent. It would be of interest to examine a panel of human NSCLC cell lines and assess the effects of this agent in the absence or presence of ATP on cell proliferation.

Reviewer #3 (Remarks to the Author): expertise in P2X7 receptor in ATP-induced purinergic signalling, drug discovery

The study reported a small-molecular-weight chemical (HEI3090) as a positive modulator of ATP-gated P2X7 receptor and demonstrated its effectiveness, particularly in combination with anti-PD-1 immune checkpoint inhibitor, in inducing antitumor effects in both Lewis lung carcinoma (LLC) transplanted and Kras-driven lung tumour mouse models. Mechanistically, the study provides evidence to suggest that HEI3090 enhances P2X7-expressing immune cells to generate interleukin (IL)-18, which in turn induces the production of interferon (IFN)-gamma by natural killer and CD4+ T cells within the tumour micro-environment (TME) and consequently upregulates MHC-I and PD-L1 expression on tumour cells. Furthermore, the study using LLC mouse model shows that HEI3090 induces strong protection against tumour re-challenge via engaging CD8+ T cells. The study is well designed, executed and presented, and the findings are scientifically and therapeutically interesting. There are several points nonetheless require clarifications to provide a clearer mechanistic insights of the role of the P2X7R in lung cancers.

Major points:

1. The authors need to provide more information of identification of HEI3090. How many compounds in the HEI's propriety chemical library were screened that led to identification and further testing of 5 promising compounds? In addition, the study claims HEI3090 to be P2X7R-specific, but there is no experimental evidence that demonstrates that it, at least at the concentration used in the study, exclusively interacts with P2X7R, but not other P2XRs.

2. The study showed that the anti-tumour effects of HEI3090 was lost in LLC-transplanted P2X7R-deficient mice and largely restored by transferring P2X7R-expressing splenocytes, pointing to preferential targeting to the P2X7R in host immune cells, rather than in tumour cells. This notion

appears to be further supported by other experiments using B16-F10 cells. However, to make this point more convincing or definitive, further experiments, for example using LLC transplanted mice are required to test whether or not treatment of tumour cells with HEI3090 results in any significant effect on tumour growth. Additionally, the authors should consider whether administration of HEI3090 in tumour-free mice before transplantation with tumour cells was also effective in producing anti-tumour effects. These studies will substantially clarify the contribution of P2X7R in tumour cells as well as host cells.

3. The study provides evidence to indicate enhanced generation of IL-18 is critical in mediating the anti-tumour effects of HEI3090. The *in vitro* results demonstrated that HEI3090 stimulated P2X7R-dependent generation of IL-18 from peritoneal macrophage cells. However, based on other experiments the authors thought that "macrophages may not be strictly necessary to mediate the anti-tumour effect of HEI3090". Thus it remains open which cell types are responsible for HEI3090-induced generation of IL-18 *in vivo* and underpin the anti-tumour effects of HEI3090. In addition, the authors have alluded that activation of NLRP3 inflammasome and caspase-1, which is required for converting IL-18 to biologically active form. The same molecular mechanism is known to trigger maturation of IL-1 β . Intriguingly, the study showed that HEI3090 was effective in stimulating generation of IL-18 but not IL-1 β . Further experiments are required to clarify whether NLRP3 and caspase-1 are indeed involved in HEI3090-induced increase in IL-18 generation. It is also important to test whether ATP induced generation of IL-1 β , and, if it is the case, whether HEI3090 enhanced ATP-induced generation of IL-1 β as shown for IL-18.

4. The study demonstrated that HEI3090 enhances the ability of anti-PD-1 in inducing anti-tumour immune response and tumour regression in both LLC-transplanted and Kras-driven lung tumour mice. The study also points to combined treatment enhanced IL-18 expression as compared to treatment with anti-PD-1 alone. It is unclear whether HEI3090 alone, enhanced the generation of IL-18, induced an anti-tumour immune effect and tumour regression in the Kras mice. Such information will help clarify whether similar or distinct molecular mechanisms drive tumour progression in the two mouse models used, which is in return critical in better understanding the anti-tumour actions of HEI3090 at the molecular and cellular levels.

5. As introduced by the authors, metastatic lung cancer is more resistant to treatment. P2X7R has been strongly implicated in regulating tumour metastasis in addition to proliferation of tumour cells. It would be highly interesting and therapeutically important to examine whether administration of HEI3090 mitigated lung tumour metastasis.

6. Finally, the study analysed the P2X7R expression in NSCLC patients and revealed that the cohort of patients with relatively high P2X7R expression was poor in survival. This seems not straightforward and clearly related to the anti-tumour effects of HEI3090 observed in lung cancer mouse models. In simple term, the high receptor expression leads to greater receptor activity, which is equivalent to that in mice after treatment with HEI3090, but favour tumour growth. Alternatively, both the P2X7R in immune cells, which is anti-tumour, and the P2X7R in tumour cells, which is pro-tumour, contribute to the pathogenesis in these patients?

Minor point:

1. P2RX7 is not the name used for the receptor or receptor protein/subunit.

Point by point response to the referees' comments

Reviewer #1 (Remarks to the Author): expertise in purinergic receptors, immune cells, cancer, immunotherapy

The authors report on a novel pharmacological modulator of P2RX7 receptor that potentiates anti-PD-1 treatment to effectively control the growth of lung tumors in an oncogene-induced NSCL and a transplantable NSCLC model leading to long lasting antitumor immune responses. The authors show that the mechanism is based on IL-18 production via P2X7 activation which leads to the production of IFN- γ by Natural Killer and CD4+ T cells within the TME. The study is interesting and has clinical potential.

We thank the reviewer for this very positive comment.

1. P2X7 activation also triggers IL-1beta production and IL-1beta was shown to cause immune escape in different solid tumors such as breast cancer (Kaplanov I et al. PNAS 2019). Did the authors study IL-1beta production?

Yes, we showed in Supplementary Figure 5D-F of the original manuscript that IL-1 β was produced and that HEI3090 does not modulate its levels. Moreover, its neutralization did not impact tumor growth in HEI3090-treated mice (Figure 4A of the original and revised manuscript). IL-1 β production in mice sera is now included in the Figure 4D of the revised manuscript.

HEI3090 does not require IL-1 β for its anti-tumoral activity and does not enhance its production *in vivo*. IL-1 β concentration in mice sera measured by ELISA.

2. P2X7 activation leads to INflammasome activation -was Nlrp3 activation analyzed, e.g. caspase-1 cleavage?

We agree that studying the NLRP3 inflammasome is important since IL-18 is known to be released upon its activation and after P2RX7's triggering by ATP. To characterize the role of the NLRP3 inflammasome in the IL-18 release by HEI3090, we used two complementary strategies. We first inhibited the NLRP3 inflammasome with the MCC950 inhibitor (described for its specificity Coll et al, Nature medicine 2015) and studied IL-1 β and IL-18 release on *ex vivo* cultured macrophages. We can see that inhibiting NLRP3 decreases drastically IL-18 and IL-1 β release from peritoneal macrophages, meaning that the NLRP3 inflammasome is involved in IL-18 release by HEI3090. This data is presented in Figure 4E of the revised manuscript.

We then studied the pro-caspase-1 cleavage, ASC and NLRP3's expression on these cells. We can see that HEI3090 increases NLRP3 and ASC's expression, and more importantly enhances pro-caspase-1 cleavage (Supplementary Figure 7 of the revised manuscript).

Altogether, these data demonstrate that HEI3090 requires the NLRP3 inflammasome and enhances the cleavage of pro-caspase-1, to increase IL-18 release.

However, IL-1 β release from these cells (Supplementary Figure 8D of the revised manuscript), as well as in mice sera (Figure 4D, right panel of the revised manuscript), is not enhanced by HEI3090. We are currently investigating why HEI3090 does not affect IL-1 β release. This will be subject to another paper dedicated to the differential regulation of IL-1 β and IL-18.

3. Figure 4F: it is questionable to quantify a cytokine by IHC. During the tissue processing most cytokines are washed out.

We agree with the reviewer. Indeed, in Figure 4C images illustrated an increased intratumor expression of IL-18 in mice treated with HEI3090. In Figure 4D, IL-18 concentration was measured by ELISA in the serum of indicated mice. We rewrote the legend to clarify this point. Since IL-18 staining and IL-18 concentrations in mice sera were coherent, we believe that the IHC are indeed representative of IL-18 presence in the TME.

4. What cell type produces IL-18 in the TME?

We agree that it is important to identify which cell is targeted by HEI3090 to produce IL-18.

Dendritic cells (DC) and macrophages are known to express P2RX7 and to be IL-18 producers. We provide new data (Figure 3B of the revised manuscript) based on adoptive transfer of WT DC into

p2rx7^{-/-} mice which clearly demonstrate that P2RX7's expressing-DC are required to mediate the antitumoral effect of HEI3090.

HEI3090 targets P2RX7-expressing dendritic cells.

One day prior to tumor cell inoculation, $1.2 \cdot 10^6$ purified DC from the spleen of WT mice were adoptively transferred by i.v. route into *p2rx7^{-/-}* mice. $5 \cdot 10^5$ LLC cells were inoculated s.c. and mice were treated daily with 2.5 mg/kg of HEI3090 or vehicle. Tumor area was measured with a caliper. At the end of the experiment, tumors were weighted, and sera was collected for ELISA.

This result suggests that the copresence of DC, live tumor cells, ATP and HEI3090 represent the perfect microenvironment where DC are primed by tumor antigens and activated by the ATP/P2RX7 axis to release IL-18. We can indeed see a slight increase in IL-18 release in sera of HEI3090-treated mice at the end of the experiment (D12). It is not that surprising not to see a major difference since mice received once WT DCs at the beginning of the experiment (D-1) and the IL-18 produced will be degraded since then. However, we can see that CD45+ cells, CD8 and NK cells in the TME show an increase in the IFN- γ /IL-10 ratio in the HEI3090-treated mice, showing that immune cells are differentiated into anti-tumor effector cells and suggesting that IL-18 produced by WT DC was efficient during the experiment. The following results are now included in the revised manuscript (Extended data and Supplementary Figure 9B).

HEI3090 targets P2RX7-expressing dendritic cells. One day prior to tumor cell inoculation, $1.2 \cdot 10^6$ purified DC from the spleen of WT mice were adoptively transferred by i.v. route into *p2rx7^{-/-}* mice. $5 \cdot 10^5$ LLC cells were inoculated s.c. and mice were treated daily with 2.5 mg/kg of HEI3090 or vehicle. At the end of the experiment, sera was collected for ELISA and tumor collected to assay cytokine production within the TME.

We also used clodronate liposomes to deplete phagocytic cells, namely macrophages and DC. In HEI3090 treated mice, the antitumoral effect is lost. This result shown in Supplementary Figure 4C confirmed that phagocytic cells mediate the antitumoral effect of HEI3090.

Three days prior to tumor cell inoculation, liposome clodronate (200 μ L) was injected ip to mice. Then mice were injected every 3 days with liposome clodronate 1h before HEI3090 treatment. $5 \cdot 10^5$ LLC cells were inoculated s.c. and mice were treated daily with 1.5 mg/kg of HEI3090 or vehicle. Tumor area was measured with a caliper. At the end of the experiment, tumors were weighted.

This experiment, combined with the observation that HEI3090 efficiently inhibit tumor growth in myeloid-specific P2RX7 deficiency (*P2rx7^{fl/fl}-LysM*) mice (Supplementary Figure 4B of the revised manuscript), demonstrates that the macrophages could participate to mediate the antitumoral effect of HEI3090, but they are not the absolutely required unlike DC (see above). These results are now discussed in the revised manuscript.

5. Figure 5D: it is hard to believe that blocking IL-18 completely blocks the P2X7 activation induces effects on IFN γ pos cells. P2X7 activation causes Nlrp3 activation and IL1beta release which is not blocked by anti IL18.

We agree that IL-1 β release is not blocked in mice that were treated with a neutralizing α -IL-18 antibody. As stated above, we have in fact shown that IL-1 β is produced in the sera of LLC-tumor bearing mice, but its production is not increased by HEI3090.

In Figure 5D we showed that the IFN- γ / IL-10 ratio was increased by HEI3090 in intratumor NK and T-CD4 cells. This increase is lost in mice that were treated with the neutralizing α -IL-18 antibody meaning that this increase is due to the production of IL-18 by HEI3090. We agree with the reviewer, neutralizing IL-18 did not abrogate IFN- γ production but only abrogated the increase of the IFN- γ / IL-10 ratio. It is reasonable to propose that the observed basal level of IFN- γ / IL-10 ratio is a consequence of IL-1beta production. We re wrote this part of the results in the revised manuscript.

6. Figure 6I: the follow-up of 3 weeks is too short.

We agree that the follow-up is short. But given the limited space in our animal unit, the mice couldn't be kept any longer. However, the results presented in Figure 6I clearly indicated that cured mice were protected. Indeed, 4 of the 6 mice studied in the cured group were still alive 26 days post re challenge with 3 of them bearing no tumors, whereas 6 of the 6 mice in the naïve group studied were dead, with death running from day 18 to 21.

7. IN the gene expression study, Figure 7S you also find IL-1beta upregulated in the P2X7 group - why should that be functionally irrelevant?

We do not believe that high levels of IL-1beta in *P2RX7* high expression group are functionally irrelevant. This cytokine certainly plays an important role in T cell mediated immunity and regulation of adaptive immune response. As requested by the editor, we rewrote this part of the result in the revised version of the manuscript.

Reviewer #2 (Remarks to the Author): expertise in lung cancer, immunotherapy, TME

This manuscript examines the effects of a potential novel inhibitor of P2RX7 as a potential cancer immunotherapy. While the studies are of potential interest, there are a number of problems with the studies that need to be addressed. In particular for many experiments there is a lack of experimental detail, such as dose of drug used or the time point examined. Specific points are addressed below.

Major points:

1. There is a significant literature suggesting that P2RX7 inhibitors may represent a therapeutic approach in cancer (Young et al *Frontiers Chem* 2018 for a review). Thus there is the possibility of opposing effects of activating this channel in cancer cells vs the TME. The data in this paper have focused on the stromal effects, but there at least needs to be some discussion of the literature supporting the use of inhibitors of P2RX7 vs activators. In that regard it would be important to know if the cancer cells used in this study express these channels.

We thank the reviewer for this constructive remark, which deserves further clarification. In the manuscript we showed that whereas tumor cells (LLC and B16-F10) expressed P2RX7 (Supplementary Figure 1 of the revised manuscript), the antitumor activity of HEI3090 depended on immune cells, since it is lost in *p2rx7*^{-/-} mice and very importantly it is restored after adoptive transfer of WT splenocytes (Figure 3A-C) and WT dendritic cells, as shown below (Figure 3B in the revised manuscript).

HEI3090 targets P2RX7-expressing dendritic cells.

One day prior to tumor cell inoculation, $1.2 \cdot 10^6$ purified DC from the spleen of WT mice were adoptively transferred by i.v. route into *p2rx7*^{-/-} mice. $5 \cdot 10^5$ LLC cells were inoculated s.c. and mice were treated daily with 2.5 mg/kg of HEI3090 or vehicle. Tumor area was measured with a caliper. At the end of the experiment, tumors were weighted, and sera was collected for ELISA.

We obtained additional data to characterize more precisely the biological activity of P2RX7 expressed by the two tumor cell lines (LLC and B16-F10) used in this study. These data are shown in the Supplementary Figure 1 of the revised manuscript.

In B16-F10 we confirmed by western blotting using the anti P2RX7's extracellular loop antibody that these cells expressed the receptor (left panel). We also observed that high doses of ATP induced an increase in the intracellular Ca^{2+} concentration (middle panel), which is inhibited when cells were pretreated with a P2RX7's antagonist (GSK1370319A). This demonstrates that P2RX7 expressed by B16-F10 cells is functional. We also analyzed the effect of ATP on cell proliferation using an assay based on BrdU incorporation. In the presence of 1mM ATP, the proliferation index is increased by 1.5-fold, indicating that P2X receptors expressed by these cells support cell proliferation. Addition of increasing doses of HEI3090 decreases ATP-induced cell proliferation, an effect likely due to the necroptotic activity of P2RX7 (as shown in Supplementary Figure 6 of the revised manuscript) and highlighting the capacity of HEI3090 to enhance P2RX7's biological activities.

B16-F10

Compared to B16F10 tumor cells, LLC expressed lower levels of P2RX7. Yet, high doses of ATP significantly increased intracellular Ca^{2+} concentrations (middle panel). In contrast with B16-F10 cells, 1mM ATP significantly reduced the proliferation index, an effect that is amplified by increasing doses of HEI3090 (right panel).

LLC

Collectively, these results indicate P2RX7 expressed by the two tumor cell lines used in this study is functional and that HEI3090 combined to ATP enhanced P2RX7's cell death activity *in vitro* (Supplementary Fig. 6A-B).

To go further, we tested in our models the inhibition of P2RX7. We injected tumor cells (LLC or B16-F10) in WT mice and treated them with the GSK1370319A compound, a well characterized P2RX7's

antagonist (Homerin G et al, 2019). Whereas HEI3090 efficiently inhibits tumor growth, we do not reproduce this inhibitory effect with the GSK compound.

A P2RX7 antagonist does not inhibit tumor growth. $5 \cdot 10^5$ B16-F10 or LLC cells were inoculated s.c. and mice were treated daily with 1.5 mg/kg of GSK1370319A or vehicle. Tumor area was measured with a caliper. At the end of the experiment tumors were weighted.

Collectively, these results demonstrate that the positive modulation of P2RX7 (here, with HEI3090) is more efficient to inhibit LLC and B16-F10 tumor growth than using the P2RX7 antagonist (GSK compound). We discussed this new data in the revised manuscript.

2. In Fig. 1 the authors have used a cell line that is engineered to overexpress P2RX7 as a screen for activators. It is likely that the levels of expression, which are not shown are supraphysiologic. Therefore it would be of interest to determine the effects of this agent in a cell line that normally expresses P2RX7.

Indeed, we have used a cell line which overexpresses P2RX7 (74% of P2RX7⁺ cells) to study HEI3090. We agree with the reviewer that it is important to show that HEI3090 is active on cells that express physiologic levels of P2RX7. In fact, we had already shown in Figure 1G that HEI3090 enhanced BzATP-induced macropore opening (increased %TO-PRO-3⁺ cells) in splenocytes isolated from WT mice. To go further, we now provide new data demonstrating that HEI3090 enhanced ATP-induced intracellular Ca²⁺ concentrations in WT splenocytes. These data are included in the revised manuscript.

HEI3090 increases calcium influx on splenocytes expressing physiologic levels of P2RX7 $5 \cdot 10^5$ splenocytes from WT (filled bars) or *p2rx7*^{-/-} (hatched bars) mice were loaded with FLUO-4-AM and stimulated with 50 μ M of ATP with or without various concentrations of HEI3090. Data shown is recorded at 1360 seconds after stimulation, from 2 independent experiments with 4 replicates. Mann Whitney test. Bars are mean \pm SEM. * $p < 0.05$, ** $p < 0.01$, *** $p < 0.001$, **** $p < 0.0001$.

In addition, there is little indication as to the specificity of this agent. Does it regulate any other ATP regulated ion channels?

We agree that studying the specificity of HEI3090 would be of interest. However, we believe that this represents a story by its own. We have nonetheless shown that HEI3090 does not increase calcium influx (a common feature of the P2X family), in *p2rx7^{-/-}* splenocytes, at least at the concentrations used (new data Figure 1).

We would like to draw the attention to the fact that even though HEI3090 could be able to modulate the activity of other P2X receptors, this will not change the conclusion of our study being that HEI3090 requires P2RX7 expression to mediate its antitumoral activity since this effect is lost in *p2rx7^{-/-}* mice, as demonstrated in Figure 3.

The authors present data indicating formation of a large pore in the setting of eATP and HEI3090 (Fig. 1D). However, there is no indication what the three tracings represent.

The 3 tracings represent Fluo-4-AM fluorescence in HEK cells stimulated with HEI3090 alone, ATP + DMSO and ATP + HEI3090 250nM. To increase the clarity, we have changed the color code in Figure 1 of the revised manuscript.

Furthermore, the data shown in Fig. 1E demonstrate a much more modest effect on large pore formation.

The effect is modest at 10 minutes post-stimulation, but it is highly reproducible ($p < 0.001$) and absent in HEK cells transfected with the empty vector (HEK-pcDNA6). This time point was chosen to ensure that the TO-PRO-3 signal is only due to the P2RX7's macropore opening and not to an increase in cell death induced by prolonged activation of P2RX7. However, the effect of HEI3090 gets more significant overtime.

3. Since much of the studies presented are in vivo, it is important to determine the dose that can be achieved. This is shown in Fig. 1F, where the maximum effective dose is 2 micromolar. However, there are no details provided as to what the administered dose was, and there are concerns regarding the relative transient nature of achieving this dose. More details need to be provided.

We agree with the reviewer, this important information is missing. Here, we administrated HEI3090 ip at a dose of 1.5 mg/kg, which corresponds to half of the highest soluble dose. This information has been added in the revised manuscript. We performed a pharmacokinetic analysis (Figure 1G), to characterize the clearance of HEI3090 and observed that after a period of 18 hr HEI3090 concentration is < 10 nM. Therefore, we have decided to inject HEI3090 each 24 hr. This was sufficient to inhibit tumor growth as shown in Figure 2 of the manuscript.

4. In Fig. 2 the authors examine the effects of HEI3090 in a KRas mouse model of lung cancer. Data shown in Pnaels E-H suggest additivity or synergy with anti-PD-1 therapy. However, there are no untreated animals. This is a particular concern, since several reports have indicate that these GEMM models are unresponsive to anti-PD-1 (McFadden et al PNAS 2016). Thus it is not clear from these data if we are just seeing a modest inhibitory effect, or additivity with anti-PD-1.

Genetically engineered mouse models, such as the KRas driven mouse model is a useful autonomous model system to assess the potential of novel mutations on tumorigenesis (Mc Fadden et al, 2016). Thanks to this mouse model, it was shown that the oncogenic RAS signaling promotes tumor immunoresistance by stabilizing PD-L1 mRNA stability (Coelho et al, 2017). This may explain why the anti-PD-1 treatment in KRAS-driven Lung cancer mouse model is inefficient (Herter-Sprie et al, 2016), as it is in immunocompetent LLC tumor mouse model (Figure2C and communication from CrownBio). In this study we used the inducible LSL *Kras*^{G12D} mouse model, in which lung tumors were induced by orotracheal instillation of an AdenoCre recombinase. In this model, tumor analysis is based on two criteria, the number of adenocarcinoma lesions within the lung and the area of the lesion (Sutherland et al, 2014). Quantification of the number of adenocarcinoma lesions in untreated mice versus α PD-1 showed a reduction, which is almost significant. Addition of HEI3090 increased the efficacy of the treatment to reach a decrease with a *P* value of 0.04. Similarly, the effect of α PD-1 on ADC lesion area is not significant, whereas the addition of HEI3090 drastically decreases the size of the lesion. Therefore, we concluded on the existence of a synergistic effect between α PD-1 and HEI3090. This data are now showed in Figure 2D of the revised manuscript.

HEI3090 synergizes with α -PD-1 to decrease the size of adenocarcinoma lesions in Kras mice

5. The characterization of T cells in the tumors shown in Fig 3 is incomplete. It would be important to see the gating strategy and to quantify both CD8 and CD4 populations.

We now added the gating strategy in the Figure 3I.

6. The authors present extensive data demonstrating that the effects of activating P2RX7 are mediated through IL18, and do not have a role for IL1beta in this pathway. In fact, no increases in IL1beta are detected. If activation of the inflammasome and caspase1 are medatied by P2RX7, one would anticipate at least seeing levels of ILbeta increase. This needs to be discussed.

We agree, we were also anticipating an IL-1 β increase due to fact that IL-18 and IL-1 β are activated by the NLRP3 inflammasome and caspase-1. We added new data to characterize the activation of NLRP3 inflammasome in response to HEI3090. To do so, we used two complementary strategies.

We first inhibited the NLRP3 inflammasome with the MCC950 inhibitor (described for its specificity Coll et al, Nature Medicine 2015) and studied IL-1 β and IL-18 release on *ex vivo* cultured macrophages. We can see that inhibiting NLRP3 decreases drastically IL-18 and IL-1 β release from peritoneal macrophages, meaning that the NLRP3 inflammasome is involved in IL-18 release by HEI3090 (Figure 4E and supplementary Figure 8D of the revised manuscript).

We then studied the pro-caspase-1 cleavage, ASC and NLRP3's expression on these cells. We can see that HEI3090 increases NLRP3 and ASC's expression, and more importantly enhances pro-caspase-1 cleavage (Supplementary Figure 7 of the revised manuscript).

Altogether, these data demonstrate that HEI3090 requires the NLRP3 inflammasome and enhances the cleavage of pro-caspase-1, to increase IL-18 release.

However, IL-1 β release from these cells (Supplementary Figure 8E) as well as in mice sera (Figure 4D, right panel) is not enhanced by HEI3090. We are currently investigating why HEI3090 does not affect IL-1 β release. This will be subject to another paper dedicated to the differential regulation of IL-1 β and IL-18. We have preliminary data (not shown) showing that IL-1 β release was enhanced by HEI3090 when the proteasome was inhibited in macrophages indicating that HEI3090 may target IL-1 β to degradation.

We also discuss this in the revised manuscript.

7. The TCGA data shown in Fig. 7 is difficult to interpret, due to the fact that these data cannot identify the cell type on which P2RX7 is expressed. The observation of poorer survival in patients with high P2RX7 expression might suggest that activators of this channel would not be an attractive therapeutic strategy.

We agree with the reviewer, it would have been really interesting to document which cell type express P2RX7, but these data are not available. P2RX7's high expression is in fact correlated with poorer survival, but we cannot know whether P2RX7's expression is on tumor cells, immune cells or

both. P2RX7's high expression is also correlated with a higher PD-L1 expression, which means that patients with higher PD-L1 levels are more eligible to immunotherapy. We have shown that P2RX7's activation by HEI3090 combined with the α -PD-1 immunotherapy decreases drastically tumors in LLC tumor-bearing mice but also in the Kras model. Therefore, we speculated that activating P2RX7 in these patients could be a promising approach. We now discuss this perspective in the revised version.

8. The models used in this study indicate a role for activation of P2RX7 in immune cells. However, this may be model dependent. It would be of interest to examine a panel of human NSCLC cell lines and assess the effects of this agent in the absence or presence of ATP on cell proliferation.

Our results show in fact the importance of P2RX7's activation in immune cells. We agree that this could be model dependent.

The expression of human P2RX7 on NSCLC cell lines was assayed by flow cytometry using the anti-conformational antibody designed by Buell. Red: control, Blue: hP2RX7 antibody

HEI3090 was screened at the NCI on 60 human tumor cell lines and showed no proliferation activity or cytotoxic effect (data not shown). In order to see if HEI3090 has an effect on human NSCLC cells lines, we selected the A549 and H1975 cells lines. We checked first the expression of P2RX7 by flow cytometry, using an antibody which recognizes the conformational form of P2RX7, in other words, the functional form of the receptor. Whereas no signal was detected on A549 cells, a very modest increase was observed in H1975, suggesting that these cells may express very low levels of functional P2RX7.

P2RX7's activation is described to be pro-proliferative, when expressed in HEK cells. We therefore tested the effect of HEI3090 on ATP-induced cell proliferation. In A549 cells, neither ATP, nor ATP in the presence of HEI3090 modulate the proliferation index. By contrast, in H1975 cells we observed that 1 mM ATP decreases by two-fold the proliferation index, interrogating on the nature of P2X receptors expressed by this cell line. However, HEI3090 does not enhance the effect of ATP.

In addition, another one of our studies on P2RX7 in LUAD patients shows that tumor cells of LUAD patients expressed a nonfunctional P2RX7 (Benzaquen et al, Theranostics, in press) whereas immune cells had a functional P2RX7.

Collectively, these data show that targeting P2RX7's expressing immune cells is an important strategy to inhibit lung tumor growth.

Reviewer #3 (Remarks to the Author): expertise in P2X7 receptor in ATP-induced purinergic signalling, drug discovery

The study reported a small-molecular-weight chemical (HEI3090) as a positive modulator of ATP-gated P2X7 receptor and demonstrated its effectiveness, particularly in combination with anti-PD-1 immune checkpoint inhibitor, in inducing antitumor effects in both Lewis lung carcinoma (LLC) transplanted and Kras-driven lung tumour mouse models. Mechanistically, the study provides evidence to suggest that HEI3090 enhances P2X7-expressing immune cells to generate interleukin (IL)-18, which in turn induces the production of interferon (IFN)- γ by natural killer and CD4+ T cells within the tumour micro-environment (TME) and consequently upregulates MHC-I and PD-L1 expression on tumour cells. Furthermore, the study using LLC mouse model shows that HEI3090 induces strong protection against tumour re-challenge via engaging CD8+ T cells. The study is well designed, executed and presented, and the findings are scientifically and therapeutically interesting.

We thank the reviewer for these positive comments.

There are several points nonetheless require clarifications to provide a clearer mechanistic insights of the role of the P2X7R in lung cancers.

Major points:

1. The authors need to provide more information of identification of HEI3090. How many compounds in the HEI's proprietary chemical library were screened that led to identification and further testing of 5 promising compounds?

One hundred and twenty compounds bearing a pyrrolidin-2-one ring from the HEI's proprietary chemical were screened for their ability to increase P2RX7-mediated intracellular TO-PRO-3 concentration during external ATP exposure and allowed to identify 5 promising compounds.

We now provide this information in the revised manuscript.

In addition, the study claims HEI3090 to be P2X7R-specific, but there is no experimental evidence that demonstrates that it, at least at the concentration used in the study, exclusively interacts with P2X7R, but not other P2XRs.

In order to characterize HEI3090's biological activity on P2RX7, we have characterized the effect of HEI3090 on 3 hallmarks of P2RX7's activation, namely increased intracellular Ca^{2+} concentration, macropore opening, and IL-18 production.

-We first used the stably transfected HEK293T cells to demonstrate that HEI3090 enhanced intracellular Ca^{2+} concentrations and macropore opening only in cells that express P2RX7 (Original manuscript, Figure 1 B to D). We then used splenocytes isolated from WT and *p2rx7*^{-/-} mice (Original manuscript, Figure 1G) to demonstrate that: (i) enhanced macropore opening (TO-PRO-3 uptake) induced by HEI3090 is still seen in endogen-P2RX7 expressing cells and that (ii) this effect is lost when splenocytes do not express P2RX7.

-We also used peritoneal macrophages isolated from *p2rx7^{-/-}* mice (Original manuscript, Figure 4E and Sup Figure 5B) to demonstrate that enhanced IL-18 production induced by HEI3090 is lost when macrophages do not express P2RX7.

-We now provide new data demonstrating that HEI3090 enhanced ATP-induced intracellular Ca²⁺ concentrations in WT splenocytes an effect that is lost when macrophages do not express P2RX7 (Figure below). These data are included in the Figure 1F of the revised manuscript.

HEI3090 increases calcium influx on splenocytes expressing physiologic levels of P2RX7 5.10^5 splenocytes from WT (filled bars) or *p2rx7^{-/-}* (hatched bars) mice were loaded with FLUO-4-AM and stimulated with 50 μM of ATP with or without various concentrations of HEI3090. Data shown is recorded at 1360 seconds after stimulation, from 2 independent experiments with 4 replicates. Mann Whitney test. Bars are mean ± SEM. *p<0.05, **p<0.01 ***p<0.001, ****p<0.0001

Collectively, these data demonstrated that HEI3090 required the expression of P2RX7 to modulate ATP-induced macropore opening, Ca²⁺ increased concentration and IL-18 secretion in immune cells. We agree that we cannot formally exclude that HEI3090 may potentially interact with other P2X receptors which was not directly evaluated here. Having not formally demonstrated the specificity of HEI3090 for P2RX7 we removed the word “specific” in the revised manuscript and rather concluded that “These results demonstrate that HEI3090 requires P2RX7 expression to be active and enhances eATP-induced P2RX7 activation”.

We would like to draw the attention to the fact that even though HEI3090 could be able to modulate the activity of other P2X receptors, this will not change the conclusion of our study being that HEI3090 requires P2RX7 expression to mediate its antitumoral activity.

We agree that characterizing the mode of action and the specificity of HEI3090 is important. However, this is a study by its own, and we started a collaboration with a biotech company who is interested by our compound.

2. The study showed that the anti-tumour effects of HEI3090 was lost in LLC-transplanted P2X7R-deficient mice and largely restored by transferring P2X7R-expressing splenocytes, pointing to preferential targeting to the P2X7R in host immune cells, rather than in tumour cells. This notion appears to be further supported by other experiments using B16-F10 cells. However, to make this point more convincing or definitive, further experiments, for example using LLC transplanted mice are required to test whether or not treatment of tumour cells with HEI3090 results in any significant effect on tumour growth.

During the process of the revision we deeper characterized P2RX7's activity in B16-F10 and LLC tumor cells (Supplementary Figure 1 of the revised manuscript).

In B16-F10 we confirmed by western blotting using the anti P2RX7's extracellular loop antibody that these cells expressed the receptor (left panel). We also observed that high doses of ATP induced an increase in the intracellular Ca^{2+} concentration (middle panel), which is inhibited when cells were pretreated with a P2RX7's antagonist (GSK1370319A). This demonstrates that P2RX7 expressed by B16-F10 cells is functional. We also analyzed the effect of ATP on cell proliferation using an assay based on BrdU incorporation. In the presence of 1mM ATP, the proliferation index is increased by 1.5-fold, indicating that P2X receptors expressed by these cells support cell proliferation. Addition of increasing doses of HEI3090 decreases ATP-induced cell proliferation, an effect likely due to the necroptotic activity of P2RX7 (as shown in Supplementary Figure 6 of the revised manuscript) and highlighting the capacity of HEI3090 to enhance P2RX7's biological activities.

Compared to B16F10 tumor cells, LLC expressed lower levels of P2RX7. Yet, high doses of ATP significantly increased intracellular Ca^{2+} concentrations (middle panel). In contrast with B16-F10 cells, 1mM ATP significantly reduced the proliferation index, an effect that is amplified by increasing doses of HEI3090 (right panel).

Collectively, these results indicate P2RX7 expressed by the two tumor cell lines used in this study is functional and that HEI3090 combined to ATP enhanced P2RX7's cell death activity *in vitro* (Supplementary Fig. 6A-B).

Since B16-F10 cells express higher levels of P2RX7, we used them to test whether or not pretreatment of tumor cells with HEI3090 results in any significant effect on tumor growth *in vivo*.

B16-F10 cells were pretreated or not with HEI3090 for 6hr prior to mice inoculation. Mice then received a daily injection of vehicle.

We show here that the pretreatment of B16-F10 cells with HEI3090 does not affect tumor growth. These results are now discussed in the revised manuscript.

Additionally, the authors should consider whether administration of HEI3090 in tumour-free mice before transplantation with tumour cells was also effective in producing anti-tumour effects. These studies will substantially clarify the contribution of P2X7R in tumour cells as well as host cells.

During the characterization of HEI3090, we showed that (1) HEI3090 needs high eATP presence to be active (Figure 1 of the original manuscript), which is present in the TME (Pellegati et al, 2008) and that (2) P2RX7-expressing immune cells are necessary for triggering anti-tumor immunity and tumor growth inhibition (Figures 3 and 4 of the original manuscript). The copresence of ATP, HEI3090, live tumor cells and immune cells represent the perfect microenvironment where immune cells can be primed by tumor antigens and activated by the ATP/P2RX7 axis to release IL-18. Tumor antigens, held by inoculated tumor cells, are important to induce a tumor-targeted immune response. Moreover, we showed that HEI3090 alone does not stimulate the production of IL-18 by macrophages *ex vivo* (See Figure 4E). We therefore hypothesized that pretreatment of mice with HEI3090 alone would not affect priming of immune cells given the absence of tumor antigens and high eATP levels. Therefore, we ethically decided not to use mice for an experiment that won't give clear cut conclusions. In the light of all the results presented in the revised manuscript, we do hope you will agree with our decision.

3. The study provides evidence to indicate enhanced generation of IL-18 is critical in mediating the anti-tumour effects of HEI3090. The *in vitro* results demonstrated that HEI3090 stimulated P2X7R-dependent generation of IL-18 from peritoneal macrophage cells. However, based on other experiments the authors thought that "macrophages may not be strictly necessary to mediate the anti-tumour effect of HEI3090". Thus it remains open which cell types are responsible for HEI3090-induced generation of IL-18 *in vivo* and underpin the anti-tumour effects of HEI3090.

We agree that it is important to identify which cell is targeted by HEI3090 to produce IL-18.

Dendritic cells (DC) and macrophages are known to express P2RX7 and to be IL-18 producers (Mutini et al, 1999; Solle et al, 2001). We therefore tested the effect of P2RX7 depletion in macrophages on HEI3090's activity using the LysMcre flox mice (supplementary Figure4B of the revised manuscript). We can in fact see that P2RX7's deletion in these mice does not impair HEI3090 anti-tumor effect.

We can conclude that macrophages are not required for HEI3090 anti-tumor effect. However, we cannot exclude the possibility of their minor involvement in HEI3090's activity.

We now provide new data based on adoptive transfer of WT DC into *p2rx7^{-/-}* mice (Figure 3B of the revised manuscript). We can see that we can restore HEI3090 anti-tumor effect. This proves that P2RX7's expressing-DCs are required for HEI3090's activity.

HEI3090 targets P2RX7-expressing dendritic cells. One day prior to tumor cell inoculation, $1.2 \cdot 10^6$ purified DC from the spleen of WT mice were adoptively transferred by i.v. route into *p2rx7^{-/-}* mice. $5 \cdot 10^5$ LLC cells were inoculated s.c. and mice were treated daily with 2.5 mg/kg of HEI3090 or vehicle. Tumor area was measured with a caliper. At the end of the experiment, tumors were weighted, and sera was collected for ELISA.

This result suggests that the copresence of DC, live tumor cells, ATP and HEI3090 represent the perfect microenvironment where DC are primed by tumor antigens and activated by the ATP/P2RX7 axis to release IL-18. We can indeed see a slight increase in IL-18 release in sera of HEI3090-treated mice at the end of the experiment (D12). It is not that surprising not to see a major difference since mice received once WT DCs at the beginning of the experiment (D-1) and the IL-18 produced will be degraded since then. However, we can see that CD45⁺ cells, CD8 and NK cells in the TME show an increase in the IFN- γ /IL-10 ratio in the HEI3090-treated mice, showing that immune cells are differentiated into anti-tumor effector cells and suggesting that IL-18 produced by WT DC was efficient during the experiment. The following results are now included in the revised manuscript (Supplementary Figure 9B of the revised manuscript).

Even though macrophages are not majorly implicated in HEI3090's effect *in vivo*, they could have a secondary role to DC in HEI3090 anti-tumor activity. Therefore, we used them as a cell type to study IL-18 release since they are known to express P2RX7 and to be IL-18 producers.

In addition, the authors have alluded that activation of NLRP3 inflammasome and caspase-1, which is required for converting IL-18 to biologically active form. The same molecular mechanism is known to trigger maturation of IL-1 β . Intriguingly, the study showed that HEI3090 was effective in stimulating generation of IL-18 but not IL-1 β . Further experiments are required to clarify whether NLRP3 and caspase-1 are indeed involved in HEI3090-induced increase in IL-18 generation.

We agree that studying the NLRP3 inflammasome is important since IL-18 is known to be released upon its activation and after P2RX7's triggering by ATP. To characterize the role of the NLRP3 inflammasome in the IL-18 release by HEI3090, we used two complementary strategies.

We first inhibited the NLRP3 inflammasome with the MCC950 inhibitor (described for its specificity Coll et al, nature medicine 2015) and studied IL-1 β and IL-18 release on *ex vivo* cultured macrophages. We can see that inhibiting NLRP3 decreases drastically IL-18 (Figure 4E of the revised manuscript) and IL-1 β release (Supplementary Figure 8D) from peritoneal macrophages, meaning that the NLRP3 inflammasome is involved in IL-18 release by HEI3090.

We then studied the pro-caspase-1 cleavage, ASC and NLRP3's expression on these cells. We can see that HEI3090 increases NLRP3 and ASC's expression, and more importantly enhances pro-caspase-1 cleavage. This data is shown in Supplementary Figure 7 of the revised manuscript.

Altogether, these data demonstrate that HEI3090 requires the NLRP3 inflammasome and enhances the cleavage of pro-caspase-1 into caspase-1, to increase IL-18 release.

HEI3090 enhances IL-18 in *ex vivo* cultured peritoneal macrophages in a NLRP3-dependant manner. $4 \cdot 10^5$ peritoneal macrophages from WT mice were primed for 4 hours with 100 ng/ml LPS and stimulated for 30 minutes with indicated doses of ATP with or without 50 μ M of HEI3090. NLRP3 was inhibited when indicated with MCC950 (1 μ M) for 1 hour prior to stimulation. **A.** IL-18 and IL-1 β quantification by ELISA. t-test Bars are mean \pm SEM, **p<0.01 **B.** Western blot of peritoneal macrophages lysates primed for 4h with LPS then stimulated with 10 μ M nigericin or 3 mM of ATP with or without 50 μ M of HEI3090. **C.** Quantification of proteins over ACTB.

However, IL-1 β release from these cells as well as in mice sera is not enhanced by HEI3090. We are currently investigating why HEI3090 does not affect IL-1 β release. This will be subject to another paper dedicated to the differential regulation of IL-1 β and IL-18 and will be discussed in the revised version of the manuscript.

It is also important to test whether ATP induced generation of IL-1 β , and, if it is the case, whether HEI3090 enhanced ATP-induced generation of IL-1 β as shown for IL-18.

We agree this is important, and indeed this was done and showed in Supplementary Figure 5 D, Figure 5E and Figure 5F. We observed that IL-1 β was produced by macrophages stimulated *ex vivo* by ATP and *in vivo* in the serum of mice inoculated with LLC tumor cells. However, because HEI3090 did not enhance the expression levels of IL-1 β , in contrast to IL-18, we decided to show this data in the supplementary figures. In the revised manuscript, the levels of IL-1 β in serum of mice are now presented in Figure 4D.

HEI3090 does not require IL-1 β for its anti-tumoral activity and does not enhance its production *in vivo*. IL-1 β concentration in mice sera measured by ELISA.

4. The study demonstrated that HEI3090 enhances the ability of anti-PD-1 in inducing anti-tumour immune response and tumour regression in both LLC-transplanted and Kras-driven lung tumour mice. The study also points to combined treatment enhanced IL-18 expression as compared to treatment with anti-PD-1 alone. It is unclear whether HEI3090 alone, enhanced the generation of IL-18, induced an anti-tumour immune effect and tumour regression in the Kras mice. Such information will help clarify whether similar or distinct molecular mechanisms drive tumour progression in the two mouse models used, which is in return critical in better understanding the anti-tumour actions of HEI3090 at the molecular and cellular levels.

We agree that it is important to show the untreated mice in this oncogenic lung tumor model and we have added this information in the revised manuscript. A new panel F is now provided. The results showed a synergistic effect between α PD1 and HEI3090 on the size of ADC lesions.

HEI3090 synergizes with α -PD-1 to decrease the size of adenocarcinoma lesions in *Kras* mice

Reviewer 3 was concerned by the fact that HEI3090 alone has not been tested in the LSL-*KRas*^{G12D} model. We had very limited number of LSL-*KRas*^{G12D} mice allowing us to compare only two groups. In the transplantable LLC tumor model, α PD-1 did not induce tumor regression; however, the combo treatment did (Figure 2A-C). In addition, the combo treatment induced a memory response (Figure 6). Furthermore, only few cancer patients achieve a response with anti-immune checkpoint administered as single-agent and, today, companies are looking for combined therapies to enhance antitumor immunity and bring a clinical benefit for patients. Therefore, it was important for us to use mice treated with α PD-1 as a control. However, we agree that testing HEI3090's efficacy in the KRAS-driven model is also important. Because of our animal facility restrictions, due to the COVID19 lockdown, adenocarcinoma was administered to 10 LSL-*KRas*^{G12D} mice. All mice in the facility were under a lot of stress during lockdown and 4 mice had to be sacrificed, according to the ethical regulations. However, the remaining mice in our experiment were analyzed and the results are shown below.

HEI3090-treated mice presented less adenocarcinoma than vehicle treated mice and the area of the lesion is smaller in HEI3090 treated mice. In agreement with what we observed in the LLC tumor mice model, IL-18 concentration in the sera of mice treated with HEI3090 tends to be increased by HEI3090 in contrast to IL-1 β . These results suggest that HEI3090 enhances the generation of IL-18 and induces tumor regression. However, given the very small number of mice remaining in this experiment, we decided to not include this result in the revised manuscript.

5. As introduced by the authors, metastatic lung cancer is more resistant to treatment. P2X7R has been strongly implicated in regulating tumour metastasis in addition to proliferation of tumour cells. It would be highly interesting and therapeutically important to examine whether administration of HEI3090 mitigated lung tumour metastasis.

We agree with the reviewer, showing that administration of HEI3090 could mitigate lung tumor metastasis would be very interesting. Indeed, we started a collaboration with Drs Laetitia Seguin and Chloé Feral to obtain the metastatic KRasG12D/p53 lung tumor model. These results will be included in an upcoming manuscript.

6. Finally, the study analysed the P2X7R expression in NSCLC patients and revealed that the cohort of patients with relatively high P2X7R expression was poor in survival. This seems not straightforward

and clearly related to the anti-tumour effects of HEI3090 observed in lung cancer mouse models. In simple term, the high receptor expression leads to greater receptor activity, which is equivalent to that in mice after treatment with HEI3090, but favour tumour growth. Alternatively, both the P2X7R in immune cells, which is anti-tumour, and the P2X7R in tumour cells, which is pro-tumour, contribute to the pathogenesis in these patients?

We thank the reviewer for this comment. We have rewritten this part of the manuscript to make it clearer. P2RX7's high expression is in fact correlated with poorer survival, but we cannot know whether P2RX7's expression is on tumor cells, immune cells or both. P2RX7's high expression is also correlated with a higher PD-L1 expression, which means that patients with higher PD-L1 levels are more eligible to immunotherapy. We have shown that P2RX7's activation by HEI3090 combined with the α -PD-1 immunotherapy decreases drastically tumors in LLC tumor-bearing mice but also in the Kras model. Therefore, we speculated that activating P2RX7 in these patients could be a promising approach.

Understanding the role of P2RX7 in the pathogenesis of LUAD patients is important. To explore this question, we focused our attention on P2RX7's functional features in *ex vivo* lung cancer samples in another study that is currently in press in Theranostics. We developed new tools that led us to demonstrate that P2RX7 is functional in leukocytes whereas it is nonfunctional in tumor cells. In addition, we showed that the expression of *P2RX7B* splice variant in tumor immune cells is associated with less infiltrated tumors in lung adenocarcinoma. Mechanistically, we observed that the differential expression of the *P2RX7B* splice variant in immune cells within tumor area correlates with the expression of a less functional P2RX7 and lower leukocytes recruitment into LUAD. Herein, we proposed that P2RX7B may participate in tumor development and therefore represent an attractive theranostic tool.

Benzaquen et al, "P2RX7B is a new theranostic marker for lung adenocarcinoma patients", Theranostics. <http://www.thno.org/ms/doc/1565/epub/48229j2.pdf>

In the next future we would like to test the effect of our molecule on P2RX7B activity.

Minor point:

1. P2RX7 is not the name used for the receptor or receptor protein/subunit.

It is true that the Uniprot nomenclature recommended to use "P2X purinoceptor 7" to design the homotrimer receptor encoded by *P2RX7* gene. On the other hand, research on the commonly used NCBI resources is based on the HUGO nomenclature and therefore utilize the term P2RX7 for both gene and protein, except that the gene is italicized. This is why we used the name P2RX7. We would like to draw attention to the fact that many other teams used the same term to design P2X purinoceptor 7. To increase the clarity of the manuscript we now indicate that P2RX7 is the same receptor than P2X7R.

REVIEWER COMMENTS

Reviewer #1 (Remarks to the Author):

The authors have answered all my comments. The manuscript has improved and I have no further comments.

Reviewer #2 (Remarks to the Author):

The study focuses on the role of P2RX7 in the tumor microenvironment. The authors have been extremely responsive to the previous critiques and have added new data to address reviewers' concerns. In particular they have thoroughly examined the potential for a role of P2RX7 in the cancer cells, performing both in vivo and in vitro experiments. An additional issue raised in the previous review, was the sensitivity of the KRas GEMM model to immunotherapy. There are conflicting reports regarding this, but the authors have clearly presented their finding for synergy of anti-PD-1 with HEI3090. Other issues have been addressed with further data, and the manuscript is greatly improved.

Reviewer #3 (Remarks to the Author):

The authors have made good efforts in revising this very interesting study, addressing many of the critical comments on the initial submission, particularly providing strong evidence to show that HEI3090 targets the P2X7 receptor expressed on dendritic cells to enhance activation of the NLRP3 inflammasome activation and maturation of IL-18. However, there are still a number of issues that need further clarification and validation.

Major points:

- 1) The revision includes new data that show the ability of HEI3090 to enhance ATP-induced calcium and dye uptake responses in mouse splenocytes expressing physiological levels of P2X7R and such responses were deficient in P2X7-KO cells. However, it is noticed that HEI3090 failed to induce a stimulatory effect on ATP-induced P2X7-mediated production of IL-1 β by macrophage (supplementary Fig. 8D). The authors need to test whether HEI3090 has a similar stimulatory effect on P2X7-mediated calcium and dye uptake in tumour cells LLC and B16-F10, and also macrophage and dendritic cells. In addition, the authors need to examine the effect on other P2X receptors, at least P2X4 which is well known to be present in P2X7-expressing cells, including immune cells. These experiments will help to clarify the in vivo action of HEI3090, particularly whether HET30390 selectively targets specific P2X7 isoform or P2X7-coupled signalling molecule rather than P2X7R itself in dendritic cells.
- 2) Ln217-230: "We showed that HEI3090 enhanced caspase-1 cleavage (Supplementary Fig. 7). ... Finally, primary peritoneal macrophages from WT mice cultured ex vivo with ATP and HEI3090 produce 1.5-fold more IL-18 than cells cultured with ATP and (Fig. 4E). ... Moreover, we showed that HEI3090 enhanced caspase-1 cleavage (Supplementary Fig. 7) meaning that HEI3090 was able to increase IL-18 production by enhancing the activation of the NLRP3 inflammasome." First of all, the authors only performed single western blotting experiment to show that HEI30390 enhanced activation of caspase-1 and NLRP3 inflammasome. Repeating these experiments and more quantitative analysis are required. Second, was there any statistically significant effect of P2X7-KO on the IL-18 level under vehicle- and HEI3090-treated condition? If not, the authors need to explain why? Finally, Fig.4E shows the effect of HEI3090 as well as MCC950 on ATP-induced IL-18 generation by macrophages. It is unclear whether ATP at the high concentrations used induced IL-18 generation? Regardless, HEI3090 has no effect on ATP-induced IL-18 generation! The authors implied dendritic cells are more important than macrophages in mediating the action of HEI3090, and therefore it is important to perform these experiments on dendritic cells to show indeed treatment with 3090 enhanced ATP-induced generation of IL-18, not IL-1 β by HEI3030, a critical step claimed by the authors.

3)ln263: "Importantly, IL-18 neutralization abrogated the increase of the IFN- γ /IL-10 ratio by CD4+ T cells and NK cells (Fig. 5F), suggesting that its production is a direct consequence of IL-18 release and signaling." However, the data shown in Fig.5F shows that IL-18 neutralization had no statistically significant effect!

Minor points

1) To present the supplementary data in a logical order. The authors discussed the data in supplementary figure 2 before supplementary figure 1.

2) To discuss the data accurately and clearly in the text, for example:

- Fig.1B-F: the key shown in Fig.1B is not applicable to this figure, and it is for Fig. 1C, E and F? Why labelling is only in the right panel, and not in the left panel?

-Supplementary Fig.2: a detailed figure legend is needed, and a scale bar on the figure

-Supplementary Fig.5: panel A is labelled with E.

-Supplementary Fig.9A: If BzATP induced no production of IFN- γ , why to test HEI3090 on BzATP-induced effect? Here, it is very interesting to test whether HEI3090 has an effect on PMA-induced generation of IFN- γ , an experiment that additionally help to clarify the specificity of HEI3090! HEI3090 induced a significant effect on CD8+ cell but not CD4+ cells, which was opposite to what shown in Fig.5D?

Ln164: "Whereas α PD-1 treatment reduced the number of ADC (Fig. 2D), ..." The p values shown in Fig.2 clearly indicate this is not case.

-ln165: "tumor burden in mice treated with the combo treatment is reduced by 60% compared to mice treated with α PD-1 alone". What did "tumor burden" refer to? Where is the evidence supporting this quantitative description?

-ln186: "...that macrophages are less implicated in HEI3090's effect in vivo (Supplementary Fig.4C)." The evidence shown by the authors supports an important role of dendritic cells, but there is no evidence to support this claim that macrophages are less implicated in mediating the in vivo action of HEI3090!

-ln204: "Our results showed that HEI3090 is not an immunogenic cell death inducer (see Extended data and Supplementary Fig. 6)." However, the data shown in supplementary Fig.6A-B clearly indicate that HEI3090 significantly enhanced BzATP-induced cell death in both LCC and B16-F10 cells? How are these in vitro results are reconciled with the in vivo data shown in supplementary Fig.6C

Reviewer #1 (Remarks to the Author):

The authors have answered all my comments. The manuscript has improved, and I have no further comments.

We thank reviewer 1 for her/his full support.

Reviewer #2 (Remarks to the Author):

The study focuses on the role of P2RX7 in the tumor microenvironment. The authors have been extremely responsive to the previous critiques and have added new data to address reviewers' concerns. In particular they have thoroughly examined the potential for a role of P2RX7 in the cancer cells, performing both in vivo and in vitro experiments. An additional issue raised in the previous review, was the sensitivity of the KRas GEMM model to immunotherapy. There are conflicting reports regarding this, but the authors have clearly presented their finding for synergy of anti-PD-1 with HEI3090. Other issues have been addressed with further data, and the manuscript is greatly improved.

We thank reviewer 2 for her/his very positive comments.

Reviewer #3 (Remarks to the Author):

The authors have made good efforts in revising this very interesting study, addressing many of the critical comments on the initial submission, particularly providing strong evidence to show that HEI3090 targets the P2X7 receptor expressed on dendritic cells to enhance activation of the NLRP3 inflammasome activation and maturation of IL-18. However, there are still a number of issues that need further clarification and validation.

Major points:

1) The revision includes new data that show the ability of HEI3090 to enhance ATP-induced calcium and dye uptake responses in mouse splenocytes expressing physiological levels of P2X7R and such responses were deficient in P2X7-KO cells. However, it is noticed that HEI3090 failed to induce a stimulatory effect on ATP-induced P2X7-mediated production of IL-1b by macrophage (supplementary Fig. 8D). The authors need to test whether HEI3090 has a similar stimulatory effect on P2X7-mediated calcium and dye uptake in tumour cells LLC and B16-F10, and also macrophage and dendritic cells.

The reviewer suggested testing the effect of HEI3090 on macrophages and dendritic cells. During the course of our study we indeed thought to do this experiment. However, the quantity of peritoneal macrophages and splenic dendritic cells is very low; tens of mice would have to be sacrificed for a single assay. To overcome this limitation (and be ethically responsible) we did this experiment directly on splenocytes (Fig.1F of the original manuscript), since immune cells isolated from spleen express P2RX7 at physiological levels and we indeed confirmed that HEI3090 enhanced both Ca²⁺ fluxes and TO-PRO-3 uptake, an effect that is lost in *p2rx7*^{-/-} mice. Moreover, splenocytes were chosen since macrophages and dendritic cells are the main cell subsets expressing high levels of P2RX7 in the spleen.

To fulfill the reviewer's suggestion, we differentiated macrophages (BMDM) and dendritic cells (BMDC) from bone marrow of wild type mice to have higher number of cells and overcome the small number of primary isolated cells. Twenty-five mice were sacrificed to set up the experimental conditions to

assay ATP-induced Ca^{2+} influx and TO-PRO-3 uptake. Despite our efforts, we were unable to evidence an effect of HEI3090 (positive or negative) on ATP-induced calcium concentration and TO-PRO-3 uptake in BMDM and we failed to see any statistically significant effect of ATP+/- HEI3090 in BMDC (Fig.1 below).

We are convinced that we have to solve technical problems and today we don't know the exact nature of these problems. For instance, it could be that ATP-induced cellular responses in BMDM and BMDC do not solely depend on P2RX7 expression, but rather on their activation and differentiation stage impacting the physiological properties of P2RX7 (Englezou P et al, Cytokine, 2015). It was also described for mouse T cells that activation and differentiation stage are more important than P2RX7 expression levels to trigger ATP-induced cellular responses (Safya H. et al, Frontiers in Immunology, 2018). Therefore, we could easily imagine that *in vitro* differentiation is different from *in vivo* differentiation.

Figure 1: ATP-induced Ca^{2+} influx and TO-PRO-3 uptake in BMDM and BMDC

A. BMDM or BMDC were loaded with Fluo4-AM in sucrose buffer for 1hr at 37°C and stimulated with ATP in the presence of increasing doses of HEI3090. Histograms represent results measured 50sec post stimulation. **B.** TO-PRO-3 uptake in response to ATP and the indicated doses of HEI3090 10min post stimulation in BMDM and BMDC. Error bars are SEM; t-test, p-value ***<0.001

However, we successfully demonstrated using BMDC that HEI3090 enhances ATP-induced IL-18 production and not IL-1 β , as asked by this reviewer (see Fig. 10 of this point by point response).

We do agree with the reviewer that the lack of IL-1 β release by macrophages and dendritic cells by HEI3090 could be linked to the fact that HEI3090 might not affect calcium influx (and subsequently potassium efflux) required for NLRP3 inflammasome activation. However, we did show, as per the reviewer request in the first reviewing process, that IL-18's release by HEI3090 required the NLRP3 inflammasome and that HEI3090 enhanced pro-caspase-1 cleavage to caspase-1 (fig.4E and Supplementary Fig.7 of the manuscript). Therefore, we are convinced that HEI3090 effectively enhanced calcium influx on these cells.

As for the release of IL-1 β , we hypothesized that HEI3090 affects its release, its processing or its stability. In fact, unlike IL-18, IL-1 β is described to be very tightly regulated by several mechanisms such as autophagy, proteasome, proteases, that could be triggered by P2RX7's activation. Our preliminary results have suggested that IL-1 β production is controlled by the proteasome. We are actually working on this particular mechanism of regulation, which will be the subject of another manuscript.

As for the effect of HEI3090 on LLC and B16-F10 cells, we showed in the manuscript (Supplementary Figs 6A and B) that HEI3090 enhances bzATP-induced cell death in both tumor cell lines. This cell death phenomenon is a consequence and a well described read out of ATP/P2RX7 signaling activation.

As per the reviewer request, we also assayed P2RX7's activation by HEI3090 on these cells. We show that HEI3090 enhanced ATP-induced dye uptake (TO-PRO-3) whereas HEI3090 alone did not induce TO-PRO-3 dye uptake. Of interest, co-stimulation with the well described P2RX7 antagonist (GSK1370319A, Homerin G et al, J Med Chem, 2020) reversed HEI3090 effect on ATP-induced increased TO-PRO-3⁺ cells, meaning that the enhanced TO-PRO-3 uptake by HEI3090 requires P2RX7.

Figure 2: HEI3090 enhances macropore opening in tumor cells in the presence of eATP

Tumor cells were loaded with TO-PRO-3 and stimulated with 500 μ M ATP alone or in the presence of HEI3090 or GSK, a P2RX7 antagonist. The TO-PRO-3⁺ cells were scored using a fluorescence microscope 15 minutes after stimulation. Error bars are SEM; t-test, p-value * <0.05 , ** <0.01

We also studied the effect of HEI3090 on ATP-induced calcium influx. Whereas ATP increases Fluo-4-AM fluorescence in LLC and B16-F10 cells, we were unable to show a statistically significant effect of HEI3090 on this cell response. However, we noticed that the P2RX7 antagonist (GSK1370319A) fully inhibits ATP-induced calcium influx.

Figure 3: HEI3090 does not affect calcium influx in tumor cells

Tumor cells were loaded with Fluo-4-AM and stimulated with 500 μM ATP alone or in the presence of HEI3090 or the P2RX7 antagonist (GSK) and the fluorescence was analyzed using a spectrophotometer. Error bars are SEM; t-test, p-value * <0.05 , *** <0.001 , **** <0.0001

We were surprised to see that HEI3090 was able to only affect one of P2RX7's activation readout, which lead us to address the nature of the different P2RX7 isoforms expressed by the tumor cell lines. Whereas both B16-F10 and LLC expressed P2RX7 at the expected size (70 kDa), we also observed the presence of shorter isoforms (Fig.4, below). We have previously shown that human P2RX7A and P2RX7B splice variants from lung cancer cells can oligomerize (Benzaquen et al, Theranostics, 2020). Whether the different mouse isoforms of the tumor lines can heterotrimerize to modulate the activity of P2RX7, remains an open question.

Figure 4: Expression of P2RX7 isoforms in cell lines

Cell lysates of corresponding cell lines were loaded on a 9% SDS-PAGE and subjected to immunoblotting with an anti P2RX7 extracellular loop antibody. The total actin (ACTB) level was monitored as a control for protein loading.

These new results show that P2RX7 expressed by LLC and B16F10 cells is functional (activated by ATP and inhibited by a classical P2RX7 antagonist) and that HEI3090 enhances ATP-induced TO-PRO-3 uptake. However, our results also show that tumor cells may express non-conventional P2RX7 isoforms (composed of full length P2RX7 and truncated P2RX7) since HEI3090 only enhanced the macropore function. At this step, we hypothesize that binding of HEI3090 favors pore formation in the tumor cell lines, as did LL37 in HEK293 cells transfected with the defective P2RX7 Δ C subunits (Di Virgilio et al, Frontiers in Pharmacology, 2018), but also impaired channel activity.

To summarize our data:

- we showed that HEI3090 enhanced ATP-induced Ca^{2+} concentration and TO-PRO-3 uptake in HEK cells expressing mP2RX7 and in wild type splenocytes (Fig. 1 of the manuscript);
- we showed that HEI3090 enhanced ATP-induced TO-PRO-3 uptake in tumor cell lines (LLC and B16-F10) (Fig. 2 of this point to point letter);
- we showed an increased production of IL-18 (but not IL-1 β) in the sera of mice treated with HEI3090, this effect is lost in *p2rx7*^{-/-} mice (Fig. 4 of the manuscript);
- we showed that HEI3090 enhanced ATP-induced IL-18 production, but not IL-1 β in peritoneal macrophages and BMDC (Fig. 4 of the manuscript and Fig.10 of this point by point letter). This effect depends on P2RX7 and on the activation of NLRP3 inflammasome;
- we showed that IL-18 is necessary to mediate HEI3090's antitumor effect (Fig.4 of the manuscript);
- we showed *in vivo* that antigen-presenting DC are required to mediate the antitumoral effect of HEI3090 (Figs 3B-C of the manuscript) and this effect is linked to an increased-IFN- γ production by NK and T cells, a read-out of IL-18's activity (Supplementary Fig.9).

Collectively these data support our conclusion that **HEI3090 enhanced the ATP/P2RX7/NLRP3/IL-18-driven antitumor immunity** and we do hope that we succeeded to convince the reviewer #3.

In addition, the authors need to examine the effect on other P2X receptors, at least P2X4 which is well known to be present in P2X7-expressing cells, including immune cells. These experiments will help to clarify the *in vivo* action of HEI3090, particularly whether HET30390 selectively targets specific P2X7 isoform or P2X7-coupled signalling molecule rather than P2X7R itself in dendritic cells.

Using splenocytes isolated from WT and *p2rx7*^{-/-} mice, we demonstrated that HEI3090 required the expression of P2RX7 to modulate ATP-induced macropore opening and Ca^{2+} concentration increase (Fig. 1F of the original manuscript).

P2RX4 has been described to cooperate with P2RX7 and we agree that it would be interesting to test whether HEI3090 targets P2RX4. We analyzed more deeply the splenocyte experiment by focusing on the initial ATP-induced calcium signal, i.e. 150 sec post stimulation, which represents P2RX7 and P2RX4 activation (Kawano et al, BBRC 2012). As shown below, we observed no effect of HEI3090 on *p2rx7*^{-/-} splenocytes, which are described to over express P2RX4 (Veinhold et al, 2010 PMID: 20405163).

Figure 5: HEI3090 does not affect calcium influx in p2rx7^{-/-} splenocytes

Splenocytes isolated from WT and p2rx7^{-/-} mice were loaded with Fluo-4AM, stimulated with 50 μ M ATP and the fluorescence was analyzed with a fluorescence spectrophotometer. Error bars are SEM from 2 independent experiments; t test.

To go further, we analyzed the effect of HEI3090 on ATP-induced calcium influx and TO-PRO-3 uptake using mouse P2RX4 overexpressing HEK cells. As expected mP2RX4 is activated by lower doses of ATP (3-10 μ M, Fig. 6 below). We then stimulated cells with 10 μ M ATP in the presence of increasing doses of HEI3090 and observed that it does not modulate TO-PRO-3 uptake, whereas the lowest dose tested (250 nM) may inhibit Ca²⁺ influx.

Figure 6: HEI3090 is not a positive modulator for ATP-induced mP2RX4-mediated TO-PRO-3 uptake and Ca²⁺ influx in HEK cells. A. Various concentrations of ATP were used to activate mP2RX4. TO-PRO-3 uptake over 200 sec was recorded. B. TO-PRO-3 uptake in response to ATP (10 μ M) and the indicated doses of HEI3090 150 sec post stimulation. C. Intracellular Ca²⁺ response was measured in Fluo4-AM loaded mP2RX4 HEK cells 50 sec post stimulation. Error bars are SEM; data from 2 independent experiments. Error bars are SEM. t-test, p-value * < 0.05, ** < 0.01, t test

Together these results demonstrate that HEI3090 is not a strong modulator of P2RX4 activity.

Since P2RX7 is activated by 10 to 100-fold more ATP than P2RX4 and P2RX4 is rapidly desensitized, which is not the case for P2RX7, we believe that the antitumor action of HEI3090 is mostly due to its effect on P2RX7. This point is now discussed in the manuscript.

2) Ln217-230: "We showed that HEI3090 enhanced caspase-1 cleavage (Supplementary Fig. 7). ... Finally, primary peritoneal macrophages from WT mice cultured ex vivo with ATP and HEI3090 produce 1.5-fold more IL-18 than cells cultured with ATP and (Fig. 4E). Moreover, we showed that HEI3090 enhanced caspase-1 cleavage (Supplementary Fig. 7) meaning that HEI3090 was able to increase IL-18 production by enhancing the activation of the NLRP3 inflammasome." First of all, the authors only performed single western blotting experiment to show that HEI30390 enhanced activation of caspase-1 and NLRP3 inflammasome. Repeating these experiments and more quantitative analysis are required.

We now repeated the experiment 3 times and edited a new Supplementary Fig.7. We confirmed our findings that HEI3090 enhances the cleavage of pro-caspase-1 to caspase-1 and confirm furthermore the requirement of NLRP3 in the release of IL-18 by HEI3090 (new fig. 4 of the manuscript).

Figure 7: HEI3090 increased eATP-induced caspase-1 cleavage in macrophages

4.10⁵ peritoneal macrophages from WT mice were primed for 4h at 37°C with 100 ng/ml LPS and then stimulated for 30 minutes with 10 μM nigericin or 3 mM ATP with HEI3090 or DMSO. Whole cell protein extracts were analyzed by western blotting by using antibodies recognizing NLRP3, ASC, pro-caspase-1 and active form caspase-1, and ACTB as a loading control. Relative band intensities of each protein were assessed to that of ACTB. Each point represents one mouse. t-test, p-value *p<0.05

Second, was there any statistically significant effect of P2X7-KO on the IL-18 level under vehicle- and HEI3090-treated condition? If not, the authors need to explain why?

Vehicle-treated WT and *p2rx7*^{-/-} mice show the same level of IL-18 meaning that P2RX7 is not the only trigger to the release of IL-18. No statistical difference is seen in IL-18 levels between vehicle and HEI3090-treated *p2rx7*^{-/-} mice meaning that P2RX7 is required for HEI3090's release of IL-18.

However, HEI3090-treated WT and *p2rx7*^{-/-} mice show indeed a significant difference in the release of IL-18. This confirms furthermore the requirement of P2RX7 for HEI3090's induced IL-18 release. We thank the reviewer for his remark and corresponding statistics have been added to the new Fig. 4D of the manuscript.

Figure 8: HEI3090-induced IL-18 production in sera of treated mice

Production of IL-18 in serum of treated mice determined by ELISA. Each point represents one mouse. t-test p-value * <0.05 , ** <0.01

Finally, Fig.4E shows the effect of HEI3090 as well as MCC950 on ATP-induced IL-18 generation by macrophages. It is unclear whether ATP at the high concentrations used induced IL-18 generation? Regardless, HEI3090 has no effect on ATP-induced IL-18 generation!

To characterize the effect of HEI3090 on macrophages, we isolated peritoneal macrophages from mice, these cells being classically used to study ATP-induced P2RX7 signaling. We now edited a new Fig. 4E showing that:

- 1) ATP induces IL-18 production in peritoneal macrophages,
- 2) HEI3090 enhanced this effect,
- 3) the NLRP3 inhibitor, MCC950, inhibits IL-18 production.

Figure 9: HEI3090 enhances ATP-IL-18's release in a NLRP3-dependant manner. Cells were primed for 4h at 37°C with 100 ng/ml LPS and then stimulated for 30 minutes as indicated. When indicated, NLRP3 inflammasome was inhibited for 1 hour with 1 μM of MCC950. Each point represents one mouse. t-test p-value * <0.05 , ** <0.01

The authors implied dendritic cells are more important than macrophages in mediating the action of HEI3090, and therefore it is important to perform these experiments on dendritic cells to show indeed treatment with 3090 enhanced ATP-induced generation of IL-18, not IL-1beta by HEI3030, a critical step claimed by the authors.

We have in fact shown *in vivo* that dendritic cells were mainly targeted by HEI3090 to inhibit tumor growth and to release IL-18, as per the increased production of IFN-γ by tumor-infiltrating immune cells (Fig. 3 and Supplementary Fig. 9B of the manuscript).

We agree with the reviewer that it is also important to show directly the release of IL-18 and IL-1 β by dendritic cells after HEI3090 stimulation. To do so, we generated bone marrow-derived dendritic cells (BMDC). As shown below, we confirmed our findings that HEI3090 enhanced ATP-induced production of IL-18 but not IL-1 β in BMDC.

Moreover, it was also important to study the involvement of the NLRP3 inflammasome in this release. The inhibition of NLRP3 with MCC950 clearly shows that IL-18 release by HEI3090 in dendritic cells is abrogated. This confirms that HEI3090's IL-18 release is NLRP3-dependant.

This experiment confirms furthermore the release of IL-18 by WT DCs in *p2rx7^{-/-}* mice (Supplementary Fig. 9B of the manuscript).

Overall, we have shown that HEI3090 inhibits tumor growth by releasing IL-18 by dendritic cells in a NLRP3-dependant manner.

Figure 10: HEI3090 enhances IL-18's release in BMDC in a NLRP3-dependant manner. BMDC were activated for 4 hours with 100 ng/ml LPS and then stimulated for 30 minutes with 3 mM ATP with or without HEI3090. When indicated, NLRP3 was inhibited for 1 hour with 1 μ M of MCC950. IL-18 and IL-1 β levels were determined by ELISA. Each point represents one mouse; t-test p-value <0.05

3)In263: "Importantly, IL-18 neutralization abrogated the increase of the IFN- γ /IL-10 ratio by CD4+ T cells and NK cells (Fig. 5F), suggesting that its production is a direct consequence of IL-18 release and signaling." However, the data shown in Fig.5F shows that IL-18 neutralization had no statistically significant effect!

Fig. 5D refers to IFN- γ /IL-10 ratio in the indicated immune cells in WT mice where HEI3090 induced an increase of the IFN- γ /IL-10 ratio.

Fig. 5F refers to IFN- γ /IL-10 ratio in the indicated immune cells in WT mice treated with neutralizing IL-18 antibody. In this figure, there is no difference in IFN- γ /IL-10 ratio (not statistic) when mice are treated with HEI3090, which means that IL-18 mediates the increase in the IFN- γ /IL-10 ratio.

Minor points

1) To present the supplementary data in a logical order. The authors discussed the data in supplementary figure 2 before supplementary figure 1.

We thank the reviewer for this comment. Indeed, tumor cell lines characterization was not presented in the manuscript. We now comment Supplementary Fig. 1 at the beginning of the results section.

2) To discuss the data accurately and clearly in the text, for example:
- Fig.1B-F: the key shown in Fig.1B is not applicable to this figure, and it is for Fig. 1C, E and F? Why labelling is only in the right panel, and not in the left panel?

For the clarity of the figure, we decided to show the full code color only once in the figure. We agree that it could be misleading. Thus, in the revised version, the concentration used in Figs 1B and 1D is now documented in the legend of each panel. We also corrected the legend of Fig. 1F.

-Supplementary Fig.2: a detailed figure legend is needed, and a scale bar on the figure

We added scale bar to the figure and a detailed figure legend.

-Supplementary Fig.5: panel A is labelled with E.

We thank you. We now corrected the figure.

-Supplementary Fig.9A: If BzATP induced no production of IFN- γ , why to test HEI3090 on BzATP-induced effect?

The goal of the experiment in Fig.9A was to see if HEI3090 was able to directly induce IFN- γ production or enhance bzATP-IFN- γ release by T cells and NK cells. Therefore, bzATP alone was used as a control to see (1) if P2RX7's activation induced IFN- γ release and to (2) compare the eventual levels of IFN- γ after BzATP+HEI3090 treatment to BzATP alone.

Here, it is very interesting to test whether HEI3090 has an effect on PMA-induced generation of IFN- γ , an experiment that additionally help to clarify the specificity of HEI3090!

In this experiment, PMA is used as a positive control in an optimal concentration to induce maximum cytokine release by the cells including IFN- γ (Marcelo P et al, 2003 PMID: 12730016). Therefore, we are not expecting to see any additional increase in the presence of HEI3090.

HEI3090 induced a significant effect on CD8⁺ cell but not CD4⁺ cells, which was opposite to what shown in Fig.5D?

In fig.5D, we observed a positive effect of HEI3090 on IFN- γ /IL-10 ratio produced by NK and CD4⁺ T cells, whereas the effect on CD8⁺ T cells was very low. This experiment was done on WT mice. To demonstrate that DC transmits the antitumor effect of HEI3090, we performed an adoptive transfer of WT DC to *p2rx7*^{-/-} mice. In this experiment, the IFN- γ /IL-10 ratio was increased in NK and CD8⁺ T cells.

The apparent discrepancies between Fig.5D and Supplementary Fig.9B might be linked to the use of mice with different backgrounds i.e. WT versus *p2rx7*^{-/-} animals. For instance, it is known that *p2rx7*^{-/-}

mice produce more IL-12 under inflammatory/infection conditions (Miller, C. et al, Plos one, 2015) than WT C57BL/6 animals. IL-12, synergistically with IL-18, induces IFN- γ production by T cells and NK cells (Nakahira, M. J Immunol, 2002). In addition, it is shown that IL-12 augments CD8⁺ T cells activation (Henry C. et al, J Immunol, 2008). Therefore, it is conceivable that under inflammatory conditions, CD8⁺ T cell responses are enhanced in *p2rx7^{-/-}* mice due to enhanced IL-12. This difference with the data of Fig.5D does not contradict our main conclusions because we have also shown that CD8⁺ T cells are important for the long term protective antitumor response (Fig.6 of the manuscript).

Ln164: "Whereas α PD-1 treatment reduced the number of ADC (Fig. 2D), ..." The p values shown in Fig.2 clearly indicate this is not case.

The reviewer is right, we only observed a tendency. We changed this sentence by "Whereas α PD-1 treatment tends to reduce the number of ADC"

-Ln165: "tumor burden in mice treated with the combo treatment is reduced by 60% compared to mice treated with α PD-1 alone". What did "tumor burden" refer to? Where is the evidence supporting this quantitative description?

Tumor burden was calculated by determining the mean of total tumor area of each lung. This information is given in the materials and methods section.

-Ln186: "...that macrophages are less implicated in HEI3090's effect *in vivo* (Supplementary Fig.4C)." The evidence shown by the authors supports an important role of dendritic cells, but there is no evidence to support this claim that macrophages are less implicated in mediating the *in vivo* action of HEI3090!

To test the involvement of macrophages in HEI3090 antitumor effect *in vivo*, we crossed the LysM-Cre mice (B6.129P2-Lys2tm1(cre)lfo) from the Jackson Laboratories with the *p2rx7^{loxP/loxP}* mice, as indicated in the extended data section. The resulting mice lack P2RX7 in the myeloid lineage e.g. monocytes/macrophages. Such mice are classically used to study the role of macrophages/monocytes in various biological responses (Shibata W et al, Gastroenterology, 2010; Mounier R et al, Cell Metabolism, 2013; Csoka B et al, The FASEB Journal, 2016). Myeloid cell conditional *p2rx7^{fl/fl}*-LysM mice were injected with LLC tumor cells and treated with HEI3090. As shown in Supplementary Fig. 4C, HEI3090 efficiently inhibits tumor growth in *p2rx7^{fl/fl}* LysM mice. We therefore believe that this result supports the notion that macrophages are less implicated than DC in HEI3090 effect *in vivo*, but we agree that macrophages may have a secondary role. This point is now addressed in the discussion section.

-Ln204: "Our results showed that HEI3090 is not an immunogenic cell death inducer (see Extended data and Supplementary Fig. 6)." However, the data shown in supplementary Fig.6A-B clearly indicate that HEI3090 significantly enhanced BzATP-induced cell death in both LCC and B16-F10 cells? How are these *in vitro* results reconciled with the *in vivo* data shown in supplementary Fig.6C

Immunogenic cell death refers to any type of death eliciting an immune response. The gold standard assay to test immunogenic cell death (ICD) is to inoculate mice with dying tumor cells and to evaluate

their ability to set up an antitumor immune response against living tumor cells given as a second challenge. Having shown that HEI3090 increased tumor cell death in the presence of bzATP (supplementary Figs 6 A and B), we then tested the hypothesis that HEI3090 may be an ICD inducer. As shown in Supplementary Fig.6C, injection of living tumor cells 7 days after the initial challenge with dead tumor cells did not protect mice from tumor development, demonstrating that the inoculated dying tumor cells were not able to trigger an antitumor immune response.

In vitro, the combination of BzATP and HEI3090 killed 60% of B16-F10 tumor cells, indicating that 40% of the cells resist to the treatment. We cannot exclude that HEI3090 kills some tumor cells (but not all) *in vivo* and the remaining resistant cells grew overtime. This hypothesis is further supported by the observation that in the presence of HEI3090 tumors grow in *p2rx7^{-/-}* mice. Therefore, even though HEI3090 is able to induce tumor cell death, our results show that HEI3090's anti-tumor activity relies essentially on P2RX7-expressing immune cells.

REVIEWERS' COMMENTS

Reviewer #3 (Remarks to the Author):

The authors have made highly appraisable efforts in addressing the comments raised by this reviewer by attempting a number of additional experiments. These additional experiments provide further insights into or clarifications of the action mechanism of this novel compound under in vitro and in vivo conditions, although not all new data can be interpreted straightforward. Therefore, it is important that the authors need to integrate these new data presented in the rebuttal letter into the manuscript as supplemental information, which gives the readers an opportunity to gain a more informed or balanced understanding of the findings reported in the study.

Point by point response

Reviewer #3 (Remarks to the Author):

The authors have made highly appraisable efforts in addressing the comments raised by this reviewer by attempting a number of additional experiments. These additional experiments provide further insights into or clarifications of the action mechanism of this novel compound under in vitro and in vivo conditions, although not all new data can be interpreted straightforward. Therefore, it is important that the authors need to integrate these new data presented in the rebuttal letter into the manuscript as supplemental information, which gives the readers an opportunity to gain a more informed or balanced understanding of the findings reported in the study.

We thank reviewer 3 for her/his full support.

As suggested, we now added in the new revised version data concerning the effect of HEI3090 on ATP-induced P2RX7's activities in LLC and B16-F10 tumor cells (new Extended Fig. 2), the effect of HEI3090 on ATP-induced IL-18 production by BMDCs (new Extended Fig. 10c) and the effect of HEI3090 on the mouse P2RX4 (Extended Fig. 11).